# Initial assessment of protein and amino acid digestive dynamics in protein-rich feedstuffs for broiler chickens

**M. Toghyani[1,2], L. R. McQuade[3], B. V. McInerney[3], A. F. Moss[4], P. H. Selle[2], S. Y. Liu[1,2]\***

**1** School of Life and Environmental Sciences, Faculty of Science, The University of Sydney, Sydney, NSW, Australia, **2** Poultry Research Foundation, The University of Sydney, Camden, NSW, Australia, **3** Australian Proteome Analysis Facility, Macquarie University, Sydney, NSW, Australia, **4** School of Environmental and Rural Science, University of New England, Armidale, NSW, Australia

\* sonia.liu@sydney.edu.au

**Data Availability Statement:** All relevant data are within the manuscript and its Supporting Information files.

## Abstract

A study evaluating apparent digestibilities of protein and amino acids and their corresponding digestion rates in four small intestinal sites in broiler chickens was completed to further investigate dietary optimisation via synchronised nutrient digestion and absorption. A total of 288 male Ross 308 broiler chickens were offered semi-purified diets with eight protein-rich feedstuffs, including; blood meal (BM), plasma protein meal (PPM), cold pressed (CCM) and expeller-pressed (ECM) canola meal, high (SBM HCP) and low (SBM LCP) crude protein soybean meals, lupins and peas. Diets were iso-caloric, iso-nitrogenous and the test ingredient was the sole source of dietary nitrogen. Each diet was offered to 6 bioassay cages with 6 birds per cage from day 21 to 28 post hatch. On day 28, all birds were euthanized and digesta samples were collected from the proximal jejunum, distal jejunum, proximal ileum and distal ileum to determine apparent protein and amino acids digestibility coefficients, digestion rates and potential digestible protein and amino acids. Dietary protein source significantly influenced energy utilisation, nitrogen retention, apparent protein (N) digestibilities, digestion rates and potential digestible protein along the small intestine. Diets containing BM and SBM LCP exhibited the highest protein digestion rate and potential digestible protein, respectively. Digestibility coefficients and disappearance rates of the majority of amino acids in four sections of the small intestine were influenced by dietary protein source (P < 0.01) and blood meal had the fastest protein digestion rate. In general, jejunal amino acid and protein digestibilities were more variable in comparison to ileal digestibilities, and the differences in protein and amino acid disappearance rates were more pronounced between types of feedstuffs than sources of similar feedstuffs.

## Introduction

There is a mandatory requirement for crude protein (CP) in broiler chicken diets ranging from 180 to 230 g/kg. When dietary energy density is not limiting, the quality and quantity of

**Funding:** The present study was funded by AgriFutures Chicken-meat (https://www.agrifutures.com.au/) PRJ-010216 "Formulating broiler diets based on protein and starch digestive dynamics" to author S.Y.L.

**Competing interests:** The authors have read the journal's policy and the authors of this paper have the following competing interests: SYL received funding from AgriFutures Chicken-meat (https://www.agrifutures.com.au/) - PRJ-010216. This does not alter our adherence to PLOS ONE policies on sharing data and materials.

dietary CP dictates birds' growth performance. The digestibilities of protein and amino acids are pivotal to the quality of the inclusion of any feedstuff in diets for chicken-meat production. However, a series of studies [1–3] have demonstrated that feed conversion efficiency is determined by the post-enteral, bilateral bioavailability of protein/amino acids and starch/glucose. The digestive dynamics of these macro-nutrients are more indicative of feed efficiency and protein deposition than the extent of digestion *per se* [4–6].

The rationale is that at least 20% of incoming dietary energy is consumed by the gastro-intestinal tract for nutrient digestion and absorption [7]. Watford et al. [8] suggested either glutamic acid/glutamine or glucose are catabolised to fuel enterocytes in the avian digestive tract. Starch is digested more rapidly than protein [9]; therefore, it was hypothesised that rapidly digestible protein such as whey protein and non-bound amino acids may present amino acids to the proximal sites of small intestine where glucose is abundant, thereby reducing amino acid catabolism and enhancing nitrogen retention and muscle protein deposition. Alternatively, the same result may be achievable via slowly digestible starch supplying glucose to the distal small intestine as an alternative energy substrate. Indeed, this hypothesis was supported by a meta-analysis of sorghum-based diets [6] and a subsequent study in wheat- and maize- based diets [10]. Given the importance of digestive dynamics to broiler growth performance, Liu and Selle [11] defined digestive dynamics of starch and protein as the digestion of starch and protein, absorption of glucose and amino acids from the gut lumen and their transition across enterocytes to enter the portal circulation. This broad definition considers the extent, rate and site of nutrient digestion along the small intestine and the bilateral, post-enteral bioavailability of glucose and amino acids. Feed conversion efficiency may be improved by rapidly digestible protein or slowly digestible starch and the impact of protein digestion is more pronounced than starch [6]. Dietary optimisation for balanced starch and protein digestion, glucose and amino acid absorption cannot be explored fully without appreciating rates of nutrient digestion in various feed ingredients. Therefore, a series of studies were completed to determine starch, protein and amino acid digestion and absorption rates in common feedstuffs used in Australia [3, 12], and the possibility of optimising dietary digestion rate *via* selecting feed ingredients and manipulating their inclusions was evaluated [2]. This initial study is one of a series in an ongoing project to determine the digestion rate of protein and amino acids in the small intestine of broiler chickens offered diets containing different protein-rich feedstuffs as the sole source of protein.

## Materials and methods

### Experimental design and diets

Eight atypical experimental diets were formulated to contain a single test ingredient as the sole protein (N) contributor. The protein-rich feedstuffs incorporated in these diets were: blood meal (BM), plasma protein meal (PPM); cold pressed (CCM) and expeller-pressed (ECM) canola meal; high (SBM HCP) and low (SBM LCP) protein soybean meal; lupins and field peas. The amino acid and mineral profiles in the test ingredients are shown in Table 1. Different inclusion rates of the test ingredients, dextrose and soy oil were used to formulate the experimental diets to be both iso-caloric and iso-nitrogenous. The compositions, calculated and analysed nutrient specifications of the experimental diets are listed in Table 2. The diets were also balanced for Ca and available P, but did not contain any non-bound (synthetic or crystalline) amino acids to avoid confounding effects. All diets were cold-pelleted through a Palmer PP330 pellet press at a conditioning temperature of 60°C with a residence time of 14 s and were then passed through a vertical cooler. A dietary marker (Celite$^{TM}$ World Minerals,

Table 1. The amino acid (g/kg) and mineral (g/kg) profile in test ingredients[1] (as-is).

| Amino Acids | SBM LCP | SBM HCP | CCM | ECM | BM | PPM | Peas | Lupins |
|---|---|---|---|---|---|---|---|---|
| His | 8.8 | 10.9 | 7.7 | 7.6 | 42.1 | 20.4 | 4.8 | 7.6 |
| Ser | 16.2 | 19.9 | 11.5 | 11.8 | 31.4 | 33.6 | 8.7 | 12.7 |
| Arg | 23.9 | 29.1 | 16.3 | 16.6 | 43.3 | 36.0 | 17.1 | 26.0 |
| Gly | 12.3 | 15.2 | 12.2 | 12.7 | 25.8 | 19.1 | 7.1 | 10.4 |
| Asp | 35.9 | 41.3 | 19.8 | 20.4 | 59.4 | 51.6 | 21.8 | 25.6 |
| Glu | 60.3 | 70.4 | 48.7 | 49.9 | 71.6 | 79.1 | 33.7 | 56.4 |
| Thr | 12.9 | 15.9 | 12.0 | 12.4 | 38.4 | 34.8 | 6.6 | 9.6 |
| Ala | 12.6 | 15.5 | 10.8 | 11.2 | 49.3 | 27.7 | 7.1 | 8.4 |
| Pro | 16.2 | 19.9 | 16.2 | 16.5 | 28.6 | 32.9 | 7.4 | 10.7 |
| Lys | 20.8 | 23.9 | 17.0 | 14.8 | 63.1 | 50.0 | 13.9 | 13.1 |
| Tyr | 9.4 | 11.6 | 6.0 | 6.1 | 27.4 | 30.5 | 2.9 | 6.5 |
| Met | 2.0 | 2.8 | 1.7 | 2.1 | 11.7 | 6.0 | 1.0 | 1.1 |
| Val | 15.6 | 19.6 | 13.9 | 14.2 | 48.9 | 39.2 | 8.7 | 10.8 |
| Ile | 15.2 | 19.1 | 11.1 | 11.4 | 28.5 | 21.3 | 8.0 | 11.4 |
| Leu | 25.6 | 32.1 | 19.3 | 20.0 | 81.4 | 58.9 | 14.0 | 19.1 |
| Phe | 17.4 | 21.8 | 11.4 | 11.7 | 50.5 | 35.7 | 9.2 | 11.2 |
| Crude protein[2] | 359 | 472 | 326 | 337 | 907 | 585 | 232 | 323 |
| Minerals | | | | | | | | |
| Ca | 2.49 | 3.20 | 6.76 | 6.57 | 0.71 | 0.66 | 0.68 | 1.83 |
| K | 17.6 | 22.6 | 11.4 | 11.9 | 1.74 | 2.50 | 10.1 | 9.23 |
| Na | 0.08 | 0.44 | 0.02 | 1.09 | 2.70 | 68.3 | 0.14 | 0.32 |
| P | 5.48 | 7.10 | 10.28 | 10.1 | 3.76 | 0.82 | 3.65 | 4.23 |

[1]SBM LCP: low crude protein soybean meal; HCP: high crude protein; CCM: cold-pressed canola meal; ECM: expeller-pressed canola meal; BM: blood meal; PPM: plasma protein meal.

[2]Protein was calculated from N concentration multiplied by the factor of 6.25.

Lompoc, CA, USA) was included at 20g/kg in diets as an inert acid insoluble ash marker in order to determine nutrient digestibility coefficients in four small intestinal sites.

## Bird management

This feeding study was approved by the Research Integrity and Ethics Administration of The University of Sydney (Project number 2016/1016). A total of 288 male Ross 308 chicks were procured from a commercial hatchery and were initially offered a wheat-based standard starter diet with 12.13 MJ/kg energy and 220 g/kg crude protein, offered to broiler chickens from 0–21 days post-hatch. At 21 days post-hatch, broiler chickens were individually identified (wing-tags), weighed and allocated into bioassay cages (6 birds per cage) on the basis of body-weights. Bird allocation was such that cage means and variations were almost identical (CV, 1.9%). Each dietary treatment was offered to six replicate cages from 21 to 28 days post-hatch. Broilers had unlimited access to water and feed under a "23-hour-on-1-hour off" lighting regime for the first three days and then under a "16-hour-on-8-hour-off" lighting regime for the remainder of the study. Room temperature was maintained at 32˚C for the first week, then gradually decreased to 22ºC by the end of the third week and maintained at the same temperature until the end of the feeding study. Body weight and feed consumption were monitored from which feed conversion ratio (FCR) was calculated. The incidence of dead or culled birds was recorded daily and their bodyweight was used to adjust FCR calculations.

**Table 2. Diet compositions and calculated nutrient specifications in experimental diets.**

| Treatment[1] (g/kg) | Blood meal | PPM | CCM | ECM | Lupins | Peas | SBM HCP | SBM LCP |
|---|---|---|---|---|---|---|---|---|
| Dextrose | 608 | 411 | 289 | 316 | 180 | 0 | 421 | 302 |
| Test ingredient | 224 | 347 | 621 | 602 | 628 | 874 | 430 | 564 |
| Soybean oil | 0 | 50 | 44.9 | 38.8 | 136.5 | 69.6 | 50 | 0 |
| Salt | 2.3 | 0 | 2.4 | 2.4 | 2.3 | 1.9 | 2.4 | 2.2 |
| Sodium Bicarbonate | 0.4 | 0 | 2.4 | 0 | 1.9 | 2.7 | 1.7 | 2.5 |
| Limestone | 14.3 | 12.7 | 8.8 | 8.8 | 13 | 16.1 | 12.5 | 10.3 |
| Dicalcium Phosphate | 17.9 | 20.7 | 7.7 | 9 | 15.3 | 12.1 | 15.2 | 19.3 |
| Choline chloride 60 | 0.8 | 0.8 | 0.8 | 0.8 | 0.8 | 0.8 | 0.8 | 0.8 |
| Premix[2] | 2.5 | 2.5 | 2.5 | 2.5 | 2.5 | 2.5 | 2.5 | 2.5 |
| Celite™ | 20 | 20 | 20 | 20 | 20 | 20 | 20 | 20 |
| Sand | 110 | 136 | 0 | 0 | 0 | 0 | 44 | 76 |
| *Calculated Nutrients* | | | | | | | | |
| AMEn (MJ/kg) | 12.8 | 12.85 | 12.85 | 12.84 | 13.21 | 12.89 | 12.85 | 12.9 |
| Crude Protein | 203 | 203 | 203 | 203 | 203 | 203 | 203 | 203 |
| Lys[3] | 16 | 16.9 | 11.2 | 10.3 | 8.8 | 12.2 | 11 | 11.3 |
| Met + Cys | 3.9 | 9.1 | 8.7 | 7.8 | 3.3 | 3.3 | 4.8 | 5.0 |
| Thr | 8.4 | 12.5 | 7.4 | 7.1 | 6.0 | 5.7 | 6.7 | 6.6 |
| Val | 14.5 | 13.3 | 8.9 | 8.8 | 6.9 | 7.1 | 8.4 | 8.3 |
| Ile | 2.5 | 7.5 | 6.9 | 6.7 | 7.2 | 6.5 | 8.2 | 8.0 |
| Ca | 9 | 9 | 9 | 9 | 9 | 9 | 9 | 9 |
| Av. P. | 4.5 | 4.5 | 4.5 | 4.5 | 4.5 | 4.5 | 4.5 | 4.5 |
| Cl | 2 | 12 | 2 | 2 | 2 | 2 | 2 | 2 |
| K | 0.2 | 1 | 7.8 | 7.5 | 5.2 | 7.2 | 9 | 11.9 |
| Na | 1.6 | 23.7 | 1.6 | 1.6 | 1.6 | 1.6 | 1.6 | 1.6 |
| Fibre | 2.2 | 3.5 | 54.7 | 62.6 | 107 | 51.1 | 13.9 | 37.3 |
| *Analysed Nutrients[4]* | | | | | | | | |
| Crude protein | 259 | 303 | 233 | 243 | 228 | 224 | 253 | 257 |
| His | 10.8 | 6.6 | 4.6 | 4.3 | 4.7 | 4.0 | 4.3 | 4.7 |
| Ser | 9.2 | 12.3 | 6.8 | 6.9 | 8.1 | 7.5 | 8.8 | 9.2 |
| Arg | 9.5 | 8.8 | 9.6 | 9.2 | 16.6 | 15.5 | 9.7 | 11.0 |
| Gly | 7.3 | 6.7 | 7.1 | 7.3 | 6.4 | 6.0 | 6.3 | 6.8 |
| Asp | 19.9 | 19.9 | 11.6 | 11.7 | 16.3 | 18.3 | 18.7 | 20.2 |
| Glu | 21.4 | 29.6 | 28.1 | 28.4 | 35.3 | 28.3 | 30.7 | 33.6 |
| Thr | 10.8 | 12.2 | 7.0 | 7.1 | 6.0 | 5.5 | 6.6 | 7.1 |
| Ala | 14.9 | 10.0 | 6.3 | 6.4 | 5.3 | 6.0 | 6.6 | 6.9 |
| Pro | 8.5 | 11.6 | 9.5 | 9.5 | 6.7 | 6.3 | 8.3 | 9.0 |
| Lys | 12.6 | 11.1 | 7.6 | 6.3 | 7.3 | 11.4 | 6.6 | 7.2 |
| Tyr | 6.1 | 8.3 | 3.7 | 3.6 | 4.9 | 3.7 | 4.7 | 5.2 |
| Met | 3.0 | 2.0 | 2.0 | 2.1 | 1.0 | 1.1 | 1.8 | 2.0 |
| Val | 14.0 | 13.3 | 8.0 | 8.1 | 6.4 | 6.9 | 7.8 | 8.3 |
| Ile | 8.4 | 7.3 | 6.4 | 6.5 | 6.7 | 6.4 | 7.7 | 8.2 |
| Leu | 24.0 | 21.4 | 11.7 | 11.8 | 11.8 | 11.5 | 13.6 | 14.6 |

(*Continued*)

**Table 2.** (Continued)

| Treatment[1] (g/kg) | Blood meal | PPM | CCM | ECM | Lupins | Peas | SBM HCP | SBM LCP |
|---|---|---|---|---|---|---|---|---|
| Phe | 14.1 | 12.0 | 6.6 | 6.6 | 6.8 | 7.6 | 8.7 | 9.3 |

[1]SBM LCP: low crude protein soybean meal; HCP: high crude protein; CCM: cold-pressed canola meal; ECM: expeller-pressed canola meal; BM: blood meal; PPM: plasma protein meal.

[2]Vitamin-trace mineral premix supplies in MIU/kg or mg/kg of diet: [MIU] retinol 12, cholecalciferol 5, [mg] tocopherol 50, menadione 3, thiamine 3, riboflavin 9, pyridoxine 5, cobalamin 0.025, niacin 50, pantothenate 18, folate 2, biotin 0.2, copper 20, iron 40, manganese 110, cobalt 0.25, iodine 1, molybdenum 2, zinc 90, selenium 0.3.

[3]digestible amino acid.

[4]analayses were conducted in duplicates.

### Sample collection and chemical analysis

Total excreta were collected from 25–27 days post-hatch from each cage to determine parameters of nutrient utilisation, including apparent metabolisable energy (AME), metabolisable energy to gross energy ratios (AME:GE), nitrogen (N) retention and N-corrected apparent metabolisable energy (AMEn). Excreta were dried in a forced-air oven at 80ºC for 24 h and the gross energy (GE) of excreta and diets were determined using an adiabatic bomb calorimeter (Parr 1281 bomb calorimeter, Parr Instruments Co., Moline, IL). The AME values of the diets were calculated on a dry matter basis from the following equation:

$$AME_{diet} = \frac{(Feed\ intake \times GE_{diet}) - (Excreta\ output \times GE_{excreta})}{(Feed\ intake)}$$

AME:GE Ratios were calculated by dividing AME by the GE of the appropriate diets. N contents of diets and excreta were determined using a nitrogen determinator (Leco Corporation, St Joseph, MI) and N retentions calculated from the following equation:

$$Retention\ (\%) = \frac{(Feed\ intake \times Nitrogen_{diet}) - (Excreta\ output \times Nitrogen_{excreta})}{(Feed\ intake \times Nitrogen_{diet})} \times 100$$

N-corrected AME (AMEn MJ/kg DM) values were calculated by correcting N retention to zero using the factor of 36.54 kJ/g N retained in the body [13].

The jejunum is reported to be the major site of glucose and amino acid absorption but the extent of nutrient digestion at the end of ileum is usually reported in the literature (Riesenfeld et al., 1980; Liu and Selle, 2015). Therefore, apparent digestibility coefficients of protein were determined in both jejunum and ileum in the present study. On day 28, the birds were euthanized (intra-venous injection of sodium pentobarbitone), and samples of digesta were taken from the proximal jejunum, distal jejunum, proximal ileum and distal ileum and pooled for each cage. The jejunum was demarcated by the end of the duodenal loop and Meckel's diverticulum and the ileum by Meckel's diverticulum and the ileo-caecal junction. Digesta was taken from the segment posterior to the respective mid-points. Intestinal segments were gently squeezed three times to minimise endogenous loss. Digesta samples from birds within a cage were pooled, homogenized, freeze-dried and ground through 0.5 mm screen. The digesta samples were weighed to determine mean retention time (MRT) and apparent digestibilities of nitrogen (N) using acid insoluble ash (AIA) as the inert dietary marker. Nitrogen and AIA concentrations were determined as outlined by Siriwan et al. [14]. Amino acid concentrations in MBM and digesta were analysed by the following procedures. Approximately 70 mg of

sample was hydrolysed in 20% HCl for 24 hours at 110°C. An internal standard (Norvaline and α amino butyric acid; Nva/AABA) was added to each sample following hydrolysis. Following a 1:25 dilution in ultra-pure water, ten microliters (10 μL) of the solution was derivatised using an AccQ-Tag Ultra Derivatization Kit (Waters Corporation, Milford, Mass. USA) following suppliers recommended procedures. The use of HCl as the hydrolysis reagent converted asparagine and glutamine to their acid forms, aspartic acid and glutamic acid, respectively. In the presence of HCl, the amino acid tryptophan was destroyed while cysteine/cystine were partially destroyed. Therefore, quantitation for these amino acids was not undertaken by this hydrolysis method. Subsequently, amino acid analysis was based on the method of Cohen [15] but adapted for use with an ACQUITY™ Ultra Performance LC (UPLC; Waters) system [16]. For elemental analysis, minerals were analysed on an ICP Emission Spectrometer (iCAP6000 Series) according to manufacturer's instructions (Thermo Electron Corporation. Waltham, M.A.).

Apparent digestibility coefficients of protein (N) and amino acids were calculated by the following equation:

$$\text{Digestibility Coefficient} = \frac{(\text{Nutrient/AIA})_{\text{diet}} - (\text{Nutrient/AIA})_{\text{digesta}}}{(\text{Nutrient/AIA})_{\text{diet}}}$$

Protein (N) and amino acids disappearance rates (g/bird/day) were deduced from feed intakes over the final phase of the feeding period from the following equation:

$$\text{Disappearance rate}_{(\text{g/bird/day})} = \text{Feed intake}_{(\text{g/bird})} \times \text{Dietary nutrient}_{(\text{g/kg})} \times \text{Digestibility coefficient}.$$

The method to predict mean retention time and digestion rates was firstly reported in Weurding et al. [17], Enting et al. [18]. In order to estimate digestion rate, the digestion time (t) was calculated from the sum of MRT determined in each intestinal segment. Mean retention time was calculated using the following equation:

$$\text{MRT (min)} = (1440 \times \text{AIA}_{\text{digesta}} \times \text{W})/(\text{FI}_{24\text{hr}} \times \text{AIA}_{\text{feed}})$$

Where $\text{AIA}_{\text{digesta}}$ is the AIA concentration in the digesta (mg/g), W is the weight of dry gut content (g), $\text{FI}_{24\text{hr}}$ is the feed intake over 24 hours before sampling (g), $\text{AIA}_{\text{feed}}$ is the AIA concentration in the feed (mg/g) and 1440 equals minutes per day.

The pattern of fractional digestibility coefficients was described by relating the digestion coefficient at each site with the digestion time (t). The curve of digestion was described by exponential model developed by Orskov and McDonald [19]:

$$\text{D}_t = \text{D}_\infty (1 - e^{-kt}) \tag{1}$$

Where $\text{D}_t$ (g/100g nitrogen) is the percentage of nitrogen that digested at time t (min), the fraction $\text{D}_\infty$ is the amount of potential digestible protein/nitrogen (asymptote) (g/100g nitrogen), digestion rate constant $k$ (per unit time, min$^{-1}$) would mean a 100% protein digestion within 1 min when it is equal to 1. Nitrogen digestion in this study was determined as apparent N digestibility, which unlike starch, is impacted by endogenous N flows.

## Statistical analysis

Experimental data were analysed as one-way ANOVA and pairwise comparisons were drawn by a student's t-test via JMP$^{®}$13.0.0 (SAS Institute Inc., JMP Software, Cary, NC). Pearson correlations were performed when considered relevant. Pre-planned orthogonal contrasts were used to determine if the parameters measured were statistically different between the two

protein meals of similar origin. Experimental units were cage means and a probability level of less than 5% was considered statistically significant.

## Results

### Nutrient content in test ingredients

The amino acid and mineral composition of the test ingredients used in this study are shown in Table 1. Glutamic acid was the most abundant in both SBM and CM samples followed by aspartic acid in both meals. The amino acid and mineral profile of CM were more consistent between the two CM sources than the SBMs. SBM HCP had higher concentrations of all the minerals measured compared to SBM LCP. Leucine was the most abundant AA in BM followed by glutamic acid and lysine, but in plasma protein meal glutamic acid content was higher than leucine. While BM had higher P concentration than PPM, the Na content in plasma protein meal was more than 25 times higher than BM. Similar to other vegetable-based protein meals, glutamic acid was the most abundant in both peas and lupins, followed by aspartic acid in peas and arginine in lupins. The Ca, Na and P content were higher in lupins than the peas.

### Growth performance

The influence of dietary treatment on growth performance and nutrient utilisation is presented in Table 3. Body weight gain and feed intake were statistically (P < 0.001) different among the diets. Birds fed the plasma meal, lupins and peas gained the lowest weight, respectively. FCR was the highest with lupins and peas, followed by blood and plasma meals. Orthogonal contrasts indicated that broilers fed BM diets had a higher body weight and feed intake than the plasma protein meal (P < 0.001). No significant differences (P > 0.05) were detected for weight gain, feed intake and FCR within different CM sources, and between lupins and

**Table 3. The influence of dietary treatment on growth performance from 21–28 days post-hatch and nutrient utilisation from 25–27 days post-hatch.**

| Ingredients | Weight gain | Feed intake | FCR | AME | AME:GE ratios | N retention (%) | AMEn (MJ/kg) |
|---|---|---|---|---|---|---|---|
| | g/bird | g/bird | | (MJ/kg) | | | |
| Blood meal (1) | 177$^c$ | 688$^b$ | 3.99 | 10.69$^c$ | 0.820$^{ab}$ | 72.36$^b$ | 9.77$^d$ |
| Plasma meal (2) | 16.3$^e$ | 360$^c$ | 3.702 | 13.17$^{ab}$ | 0.915$^a$ | 85.92$^a$ | 12.60$^{ab}$ |
| CCM (3) | 607$^{ab}$ | 1010$^a$ | 1.663 | 12.09$^{abc}$ | 0.693$^c$ | 61.66$^{bc}$ | 10.70$^{cd}$ |
| ECM (4) | 577$^{ab}$ | 1046$^a$ | 1.816 | 12.52$^{bc}$ | 0.732$^{bc}$ | 56.92$^c$ | 11.16$^c$ |
| Lupins (5) | 55$^{de}$ | 628$^b$ | 11.63 | 13.74$^a$ | 0.733$^{bc}$ | 52.66$^{cd}$ | 13.11$^a$ |
| Peas (6) | 80$^d$ | 720$^b$ | 9.053 | 11.93$^{bc}$ | 0.693$^c$ | 44.26$^d$ | 11.28$^{bc}$ |
| SBM HCP (7) | 553$^b$ | 1106$^a$ | 2.001 | 12.01$^{bc}$ | 0.793$^{bc}$ | 55.25$^{cd}$ | 10.57$^{cd}$ |
| SBM LCP (8) | 624$^a$ | 1055$^a$ | 1.692 | 11.90$^{bc}$ | 0.755$^{bc}$ | 52.25$^{cd}$ | 10.55$^{cd}$ |
| *SEM* | *13.81* | *34.63* | *2.635* | *0.376* | *0.025* | *2.711* | *0.302* |
| P-value | < .001 | < .001 | 0.066 | < .001 | < .001 | < .001 | < .001 |
| Pre-planned orthogonal contrast Probabilities | | | | | | | |
| Diet 1 *vs*. 2 | 0.001 | 0.001 | 0.938 | 0.001 | 0.013 | 0.001 | 0.001 |
| Diet 3 *vs*. 4 | 0.123 | 0.475 | 0.967 | 0.427 | 0.289 | 0.224 | 0.291 |
| Diet 5 *vs*. 6 | 0.216 | 0.065 | 0.493 | 0.001 | 0.271 | 0.034 | 0.001 |
| Diet 7 *vs*. 8 | 0.008 | 0.303 | 0.934 | 0.828 | 0.302 | 0.439 | 0.963 |

[a-d]Means within columns not sharing a common suffix are significantly different at P ≤ 0.05.

[1]SBM LCP: low crude protein soybean meal; HCP: high crude protein; CCM: cold-pressed canola meal; ECM: expeller-pressed canola meal.

peas. Birds fed SBM LCP diet had higher weight gain than the birds fed SBM HCP diet
(P < 0.01).

Despite the experimental diets being formulated to be isocaloric and isonitrogenous, the
energy utilisation and nitrogen retention determined from d 25–27 was significantly
(P < 0.001) different among treatments. The highest AME and AMEn were found with lupins
and the lowest with blood meal diets, but the highest AME: GE ratio was calculated with
plasma protein meal, and both CCM and peas showed the lowest ratios. Broilers fed the plasma
meal diet retained the most N, while broilers offered the pea diet had the lowest N retention.
There were significant differences between blood and plasma protein meals (P < 0.01) for
both energy utilization and nitrogen retention, with broilers fed the PPM diets showing higher
values than BM. The AME, AMEn, and N retention values were all higher in broilers fed lupins
diet than peas (P < 0.05). Nutrient utilization values were statistically similar between the two
CM sources and the two SBM sources (P > 0.05).

## Protein digestion along the small intestine

The influence of dietary treatment on protein digestive dynamics in broiler chickens at 27 days
post-hatch is shown in Table 4. Broiler chickens offered the various diets had different appar-
ent digestibility of protein (N) in four different sites of the intestine (P < 0.01), protein diges-
tion rate and potential digestible protein (P < 0.05). In the proximal jejunum, BM had the
highest apparent protein digestibility and this was also reflected in its highest protein digestion
rate (P = 0.033). In proximal and distal ileum, broiler chickens offered the PPM diet had the
highest N digestibility. Orthogonal contrasts showed that N digestibility in proximal jejunum

**Table 4. The influence of dietary treatment[1] on apparent digestibility of protein (N), potential digestible protein (N), protein (N) digestion rate and apparent pro-
tein (N) disappearance rate (g/bird/day) in broiler chickens at 27 days post-hatch.**

| Ingredients | Apparent digestibility coefficients | | | | Protein digestion rate | Potential digestible protein | Disappearance rate | | | |
|---|---|---|---|---|---|---|---|---|---|---|
| | Proximal jejunum | Distal jejunum | Proximal ileum | Distal ileum | (min$^{-1}$) | (g/g) | Proximal jejunum | Distal jejunum | Proximal ileum | Distal ileum |
| Blood meal (1) | 0.568$^a$ | 0.516$^b$ | 0.717$^{bcde}$ | 0.761$^{bcd}$ | 0.124$^a$ | 0.678$^b$ | 14.46$^{ab}$ | 13.19$^b$ | 18.18$^{cd}$ | 19.60$^{cd}$ |
| Plasma meal (2)[2] | 0.256$^b$ | 0.486$^b$ | 0.892$^a$ | 0.937$^a$ | - | - | 4.80$^b$ | - | 16.64$^d$ | 17.49$^{cd}$ |
| CCM (3) | 0.229$^b$ | 0.485$^b$ | 0.619$^e$ | 0.684$^d$ | 0.023$^{bc}$ | 0.771$^b$ | 7.74$^b$ | 16.41$^{ab}$ | 20.80$^{bcd}$ | 22.98$^{bc}$ |
| ECM (4) | 0.246$^b$ | 0.470$^b$ | 0.627$^{de}$ | 0.704$^{cd}$ | 0.015$^c$ | 0.859$^{ab}$ | 8.90$^b$ | 17.11$^{ab}$ | 22.81$^{bc}$ | 25.62$^{ab}$ |
| Lupins (5) | 0.446$^{ab}$ | 0.771$^a$ | 0.826$^{ab}$ | 0.834$^b$ | 0.062$^b$ | 0.851$^{ab}$ | 9.13$^b$ | 16.91$^{ab}$ | 16.96$^d$ | 17.08$^d$ |
| Peas (6) | 0.400$^{ab}$ | 0.639$^{ab}$ | 0.788$^{abc}$ | 0.791$^b$ | 0.051$^{bc}$ | 0.841$^{ab}$ | 8.11$^b$ | 15.51$^b$ | 18.15$^{cd}$ | 18.27$^{cd}$ |
| SBM HCP (7) | 0.460$^{ab}$ | 0.564$^{ab}$ | 0.762$^{abcd}$ | 0.765$^{bcd}$ | 0.066$^b$ | 0.804$^{ab}$ | 18.30$^a$ | 22.53$^a$ | 30.54$^a$ | 30.65$^a$ |
| SBM LCP (8) | 0.277$^b$ | 0.586$^{ab}$ | 0.678$^{cde}$ | 0.773$^{bc}$ | 0.029$^{bc}$ | 0.921$^a$ | 10.55$^b$ | 22.72$^a$ | 26.24$^{ab}$ | 29.98$^a$ |
| *SEM* | *0.074* | *0.045* | *0.031* | *0.030* | *0.021* | *0.049* | *2.815* | *1.468* | *1.274* | *1.303* |
| P-value | 0.004 | 0.002 | < .001 | < .001 | 0.033 | 0.016 | 0.035 | 0.002 | < .001 | < .001 |
| Pre-planned orthogonal contrast Probabilities | | | | | | | | | | |
| Diet 1 *vs.* 2 | 0.068 | 0.642 | 0.642 | < .001 | N/A | N/A | 0.134 | N/A | 0.437 | 0.300 |
| Diet 3 *vs.* 4 | 0.851 | 0.072 | 0.860 | 0.566 | 0.771 | 0.216 | 0.732 | 0.927 | 0.258 | 0.148 |
| Diet 5 *vs.* 6 | 0.622 | 0.802 | 0.389 | 0.238 | 0.702 | 0.906 | 0.763 | 0.518 | 0.499 | 0.509 |
| Diet 7 *vs.* 8 | 0.046 | 0.718 | 0.064 | 0.818 | 0.202 | 0.102 | 0.027 | 0.928 | 0.018 | 0.714 |

[a-e] Means within columns not sharing a common suffix are significantly different at P ≤ 0.05.

[1] SBM LCP: low crude protein soybean meal; HCP: high crude protein; CCM: cold-pressed canola meal; ECM: expeller-pressed canola meal.

[2] Broiler chickens offered Diet 2 generated very low quantity of digesta and mean retention time was not determined.

tended (P = 0.068) to be higher with BM than PPM, but in distal ileum the coefficient was higher with PPM (P < 0.001). For canola meals, broiler chickens offered the ECM diets had almost two times higher protein (N) digestibility in the distal ileum than birds offered the CCM diet (P = 0.001). Lupins also had a higher protein digestibility at the distal jejunum than peas (P < 0.05). Broiler chickens offered SBM HCP generated a higher protein digestibility in proximal jejunum (P < 0.05) in comparison to birds offered SBM LCP. The highest and lowest protein digestion rate (P < 0.05) were determined for BM and ECM diets, respectively. Potential digestible protein was calculated to be the highest in SBM LCP diet, while the lowest value was observed with BM (P < 0.05). Orthogonal contrasts did not reveal any statistical differences (P > 0.05) for protein digestion rate and potential digestible protein between diets containing similar type of protein-rich ingredients.

Apparent protein (N) disappearance rate was significantly different among experimental diets in all four sections of the small intestine (P < 0.05). SBM HCP diet consistently showed the highest disappearance rate in all four sites of small intestine. Broiler chickens offered the SBM HCP diet generated higher protein disappearance rate in the proximal jejunum and ileum than chickens offered the SBM LCP diet (P < 0.05). No other significant differences were observed in broiler chickens offered diets containing similar type of feed ingredients.

## Amino acid digestibilities along the small intestine

Table 5 summarises the effect of dietary treatments on apparent digestibility of amino acids in the proximal jejunum. The quantity of digesta samples collected from birds in the PPM diet was insufficient for amino acids analysis; therefore, no amino acid digestibility results were reported in broiler chickens offered the PPM diet. The digestibility coefficients for all the amino acids analysed were significantly different across the dietary treatments (P < 0.01). However, the differences observed did not reach statistical significance (P > 0.05) for broiler chickens offered diets containing similar type of feed ingredients with the exception of His, Pro and Tyr digestibility coefficients where birds offered lupins had higher digestibilities than broilers offered peas (P < 0.05). The effect of dietary treatments on apparent amino acid disappearance rate in the proximal jejunum is presented in Table 6. The disappearance rate of all amino acids, except Gly, was significantly influenced by dietary protein source (P < 0.05). Broiler chickens offered both blood meal and SBM HCP generated the highest disappearance rate for all the amino acids. Lys disappearance rate was higher in CCM compared to ECM (P = 0.018). Other amino acids disappearance rate was not statistically different between diets containing similar type of ingredients (P > 0.05).

The effect of dietary treatments on apparent amino acid digestibilities in the distal jejunum is presented in Table 7. Dietary treatments significantly affected the apparent digestibility of all amino acids, except Met, in the distal jejunum (P < 0.001). Diets with peas and lupins exhibited the highest digestibility values for all the amino acids compared to the other feed ingredients. However, orthogonal contrasts did not show any significant differences between diets with similar type of feed ingredients (P > 0.05). Table 8 reported apparent amino acid disappearance rate in the distal jejunum. There were significant treatment differences between apparent disappearance rates of various feed ingredients for all the amino acids except Ile and Gly. Lys disappearance rate was calculated to be higher in peas than lupins (P < 0.001). But the values for other amino acids did not differ statistically within diets containing similar type of ingredients (P > 0.05).

The influence of feed ingredients on apparent digestibilities and disappearance rates of amino acids in the proximal ileum are shown in Tables 9 and 10. The apparent digestibility coefficient of all the amino acids, except Met, determined in proximal ileum was statistically

**Table 5. The influence of dietary treatments[1] on apparent digestibility coefficients of amino acids in the proximal jejunum of broiler chickens at 27 days post-hatch.**

| Ingredients[2] | Essential amino acids | | | | | | | | | | Non-essential amino acids | | | | | | |
|---|---|---|---|---|---|---|---|---|---|---|---|---|---|---|---|---|---|
| | Arg | His | Ile | Leu | Lys | Met | Phe | Thr | Val | Ala | Asp | Glu | Gly | Pro | Ser | Tyr | Average |
| Blood meal (1) | 0.548^abc | 0.719^a | 0.590^a | 0.662^a | 0.597^a | 0.589^a | 0.654^a | 0.642^a | 0.627^a | 0.684^a | 0.640^a | 0.574^ab | 0.601^a | 0.609a | 0.636a | 0.140^a | 0.595^a |
| CCM (3) | 0.310^c | 0.332^bc | 0.119^b | 0.190^b | 0.008^b | 0.402^a | 0.210^b | 0.142^b | 0.149^b | 0.171^b | 0.163^bc | 0.340^bc | 0.190^bc | 0.253^bc | 0.116^b | -1.061^c | 0.127^b |
| ECM (4) | 0.240^c | 0.230^c | 0.090^b | 0.171^b | -0.113^b | 0.413^a | 0.146^b | 0.087^b | 0.099^b | 0.144^b | 0.087^c | 0.292^c | 0.124^c | 0.168^c | 0.089^b | -0.984^bc | 0.080^b |
| Lupins (5) | 0.722^a | 0.597^ab | 0.410^ab | 0.449^ab | 0.309^ab | -0.183^bc | 0.448^ab | 0.343^ab | 0.322^ab | 0.308^ab | 0.518^a | 0.677^a | 0.443^ab | 0.491^ab | 0.486^a | -0.062^a | 0.392^ab |
| Peas (6) | 0.640^ab | 0.408^abc | 0.252^ab | 0.306^ab | 0.496^a | -0.323^c | 0.344^ab | 0.186^b | 0.231^ab | 0.271^b | 0.481^a | 0.546^ab | 0.289^abc | 0.313^abc | 0.345^ab | -0.590^abc | 0.262^ab |
| SBM HCP (7) | 0.368^bc | 0.421^abc | 0.329^ab | 0.375^ab | 0.113^ab | 0.292^ab | 0.393^ab | 0.296^ab | 0.298^ab | 0.327^ab | 0.426^ab | 0.490^abc | 0.335^abc | 0.410^abc | 0.420^a | -0.283^ab | 0.313^ab |
| SBM LCP (8) | 0.277^c | 0.310^c | 0.197^b | 0.259^b | -0.060^b | 0.171^ab | 0.269^b | 0.163^b | 0.161b | 0.206^b | 0.304^abc | 0.382^bc | 0.219^bc | 0.288^bc | 0.291^ab | -0.509^abc | 0.183^b |
| SEM | 0.062 | 0.060 | 0.073 | 0.070 | 0.098 | 0.107 | 0.069 | 0.077 | 0.074 | 0.071 | 0.061 | 0.053 | 0.064 | 0.058 | 0.063 | 0.169 | 0.079 |
| P-value | <.001 | 0.004 | 0.003 | 0.003 | 0.001 | <.001 | 0.002 | 0.004 | 0.006 | 0.005 | <.001 | 0.002 | 0.002 | 0.001 | <.001 | 0.003 | 0.005 |
| Pre-planned orthogonal contrast Probabilities | | | | | | | | | | | | | | | | | |
| Diet 3 vs. 4 | 0.436 | 0.242 | 0.777 | 0.847 | 0.379 | 0.937 | 0.506 | 0.613 | 0.630 | 0.792 | 0.394 | 0.516 | 0.474 | 0.306 | 0.769 | 0.739 | 0.659 |
| Diet 5 vs. 6 | 0.384 | 0.041 | 0.148 | 0.172 | 0.196 | 0.373 | 0.303 | 0.172 | 0.405 | 0.735 | 0.685 | 0.098 | 0.117 | 0.047 | 0.144 | 0.035 | 0.247 |
| Diet 7 vs. 8 | 0.312 | 0.204 | 0.200 | 0.248 | 0.209 | 0.419 | 0.200 | 0.218 | 0.193 | 0.246 | 0.173 | 0.149 | 0.208 | 0.151 | 0.162 | 0.332 | 0.225 |

a-c Means within columns not sharing a common suffix are significantly different at P ≤ 0.05.

[1] SBM LCP: low crude protein soybean meal; HCP: high crude protein; CCM: cold-pressed canola meal; ECM: expeller-pressed canola meal.

[2] Broiler chickens offered diet 2 did not generate enough digesta content for completing amino acid analyses in Table 5–13.

**Table 6. The influence of dietary treatments[1] on disappearance rate (g/bird/day) of amino acids by the proximal jejunum of broiler chickens at 27 days post-hatch.**

| Ingredients | Essential amino acids | | | | | | | | | Non-essential amino acids | | | | | | | |
|---|---|---|---|---|---|---|---|---|---|---|---|---|---|---|---|---|---|
| | Arg | His | Ile | Leu | Lys | Met | Phe | Thr | Val | Ala | Asp | Glu | Gly | Pro | Ser | Tyr | Average[2] |
| Blood meal (1) | 0.506[ab] | 0.770[a] | 0.501[a] | 1.612[a] | 0.752[a] | 0.173[a] | 0.905[a] | 0.715[a] | 0.909[a] | 1.118[a] | 1.290[a] | 1.225[b] | 0.505 | 0.536[ab] | 0.616[a] | 0.082[a] | 12.22[a] |
| CCM (3) | 0.422[b] | 0.216[b] | 0.111[b] | 0.326[b] | 0.009[b] | 0.118[ab] | 0.196[bc] | 0.149[b] | 0.178[b] | 0.172[b] | 0.278[bc] | 1.379[b] | 0.226 | 0.360[ab] | 0.121[b] | -0.554[b] | 3.71[b] |
| ECM (4) | 0.323[b] | 0.147[b] | 0.088[b] | 0.307[b] | -0.105[b] | 0.129[ab] | 0.143[c] | 0.095[b] | 0.124[b] | 0.151[b] | 0.155[c] | 1.237[b] | 0.155 | 0.248[b] | 0.096[b] | -0.5170[b] | 1.78[b] |
| Lupins (5) | 1.058[a] | 0.253[b] | 0.250[ab] | 0.486[b] | 0.201[b] | -0.017[c] | 0.269[bc] | 0.189[b] | 0.190[b] | 0.159[b] | 0.773[abc] | 2.159[ab] | 0.294 | 0.310[ab] | 0.374[ab] | -0.028[ab] | 6.92[ab] |
| Peas (6) | 1.001[a] | 0.168[b] | 0.172[b] | 0.373[b] | 0.587[ab] | -0.034[c] | 0.266[bc] | 0.112[b] | 0.175[b] | 0.186[b] | 0.923[ab] | 1.597[ab] | 0.206 | 0.211[b] | 0.282[ab] | -0.214[ab] | 6.02[ab] |
| SBM HCP (7) | 0.543[ab] | 0.281[b] | 0.399[a] | 0.811[b] | 0.113[b] | 0.080[ab] | 0.526[ab] | 0.315[ab] | 0.375[b] | 0.367[b] | 1.263[a] | 2.259[a] | 0.381 | 0.556[ab] | 0.615[a] | -0.208[ab] | 8.78[ab] |
| SBM LCP (8) | 0.441[b] | 0.213[b] | 0.241[ab] | 0.570[b] | -0.071[b] | 0.048[bc] | 0.364[bc] | 0.174[b] | 0.202[b] | 0.231[b] | 0.926[ab] | 1.917[ab] | 0.255 | 0.400[ab] | 0.423[ab] | -0.392[b] | 5.95[ab] |
| *SEM* | *0.102* | *0.044* | *0.082* | *0.135* | *0.104* | *0.026* | *0.085* | *0.075* | *0.089* | *0.076* | *0.174* | *0.243* | *0.071* | *0.082* | *0.088* | *0.103* | *1.607* |
| P-value | 0.002 | < .001 | 0.026 | < .001 | < .001 | < .001 | < .001 | 0.001 | 0.003 | < .001 | 0.001 | 0.017 | 0.104 | 0.036 | 0.004 | 0.002 | 0.016 |
| Pre-planned orthogonal contrast Probabilities | | | | | | | | | | | | | | | | | |
| Diet 3 *vs.* 4 | 0.504 | 0.236 | 0.838 | 0.924 | 0.442 | 0.636 | 0.640 | 0.621 | 0.662 | 0.875 | 0.595 | 0.684 | 0.495 | 0.300 | 0.833 | 0.592 | 0.668 |
| Diet 5 *vs.* 6 | 0.709 | 0.160 | 0.507 | 0.577 | 0.018 | 0.648 | 0.980 | 0.495 | 0.522 | 0.808 | 0.537 | 0.130 | 0.412 | 0.382 | 0.450 | 0.765 | 0.688 |
| Diet 7 *vs.* 8 | 0.485 | 0.243 | 0.160 | 0.216 | 0.222 | 0.258 | 0.159 | 0.196 | 0.164 | 0.205 | 0.151 | 0.209 | 0.219 | 0.153 | 0.105 | 0.214 | 0.195 |

[a-c] Means within columns not sharing a common suffix are significantly different at P ≤ 0.05

[1] SBM LCP: low crude protein soybean meal; HCP: high crude protein; CCM: cold-pressed canola meal; ECM: expeller-pressed canola meal

[2] Average: the sum of individual amino acids

(P < 0.01) different among the dietary treatments. Diets containing lupins and peas consistently had higher digestibility coefficients for all the amino acids compared to the other dietary treatments. However, there was no significant (P > 0.05) orthogonal contrasts between the diets containing similar type of feed ingredients for all amino acids. The disappearance rates of all the amino acids were significantly (P < 0.001) influenced by dietary protein source. Both SBM sources recorded the highest disappearance rate for the majority of the amino acids, also reflected in higher average amino acid disappearance rates. Similar to its disappearance rate in the distal jejunum, Lys disappearance rate was statistically (P = 0.007) higher in peas than lupins.

The influence of feed ingredients on apparent digestibilities and disappearance rates of amino acids in the distal ileum are shown in Tables 11 and 12. Consistent with proximal ileal digestibility coefficient for amino acids, distal ileal amino acid digestibilities also followed similar pattern except Ala, which is significantly different among the dietary treatments (P < 0.001). Peas and lupins had the highest average amino acid digestibility coefficients, followed by BM and SBM. There were no significant differences between diets containing similar type of ingredients except Lys digestibility (P < 0.01) which was significantly higher in CCM than ECM. The disappearance rates of all the amino acids in distal ileum were significantly influenced by dietary protein source (P < 0.001). Broiler chickens offered soybean meals had the highest average amino acid disappearance rate, followed by birds offered blood meal and canola meal. Orthogonal contrasts indicated a significant difference for Lys disappearance rate between lupins and peas (P = 0.001), and canola meal sources (P = 0.006), with peas and CCM showing higher rate than their counterparts. The disappearance rate of Ala (P = 0.035) and Asp (P = 0.006) was higher in peas than lupins.

Table 13 present the potential digestible amino acid. The potential digestible amino acids content of the experimental diets was not significantly influenced by dietary protein source

**Table 7. The influence of dietary treatments[1] on apparent digestibility of amino acids in the distal jejunum of broiler chickens at 27 days post-hatch.**

| Ingredients | Essential amino acids | | | | | | | | | Non-essential amino acids | | | | | | | |
| --- | --- | --- | --- | --- | --- | --- | --- | --- | --- | --- | --- | --- | --- | --- | --- | --- | --- |
| | Arg | His | Ile | Leu | Lys | Met | Phe | Thr | Val | Ala | Asp | Glu | Gly | Pro | Ser | Tyr | Average |
| Blood meal (1) | 0.508[b] | 0.617[bc] | 0.570[abc] | 0.593[abc] | 0.511[bc] | 0.610 | 0.580[abc] | 0.584[abc] | 0.555[ab] | 0.607[ab] | 0.587[ab] | 0.539[b] | 0.561[abc] | 0.558[abc] | 0.612[ab] | -0.044[bc] | 0.534[abc] |
| CCM (3) | 0.596[b] | 0.585[bc] | 0.405[c] | 0.471[c] | 0.339[c] | 0.678 | 0.482[c] | 0.387[cd] | 0.421[b] | 0.454[b] | 0.419[bc] | 0.592[b] | 0.453[c] | 0.476[c] | 0.378c | -0.373[c] | 0.423[c] |
| ECM (4) | 0.564[b] | 0.498[c] | 0.401[c] | 0.470[c] | 0.234[c] | 0.633 | 0.471[c] | 0.343[d] | 0.398[b] | 0.435[b] | 0.349[c] | 0.549[b] | 0.380[c] | 0.395[c] | 0.366[c] | -0.365[c] | 0.383[c] |
| Lupins (5) | 0.881[a] | 0.807[a] | 0.718[a] | 0.747[a] | 0.702[a] | 0.569 | 0.746[a] | 0.675[a] | 0.674[a] | 0.668[a] | 0.755[a] | 0.845[a] | 0.738[a] | 0.744[a] | 0.750[a] | 0.543[a] | 0.723[a] |
| Peas (6) | 0.850[a] | 0.728[ab] | 0.671[ab] | 0.705[a] | 0.793[a] | 0.559 | 0.717[ab] | 0.629[ab] | 0.655[a] | 0.685[a] | 0.750[a] | 0.802[a] | 0.670[ab] | 0.671[ab] | 0.689[ab] | 0.347[ab] | 0.683[ab] |
| SBM HCP (7) | 0.532[b] | 0.567[bc] | 0.479[bc] | 0.515[bc] | 0.330[c] | 0.566 | 0.526[bc] | 0.447[bcd] | 0.460[ab] | 0.487[ab] | 0.504[bc] | 0.584[b] | 0.478[bc] | 0.526[bc] | 0.542[bc] | 0.031[bc] | 0.473[bc] |
| SBM LCP (8) | 0.553[b] | 0.552[c] | 0.450[c] | 0.484[c] | 0.331[c] | 0.552 | 0.494[c] | 0.411[cd] | 0.422[b] | 0.448[b] | 0.488[bc] | 0.569[b] | 0.448[c] | 0.503[bc] | 0.496[bc] | 0.048[bc] | 0.453[abc] |
| *SEM* | *0.044* | *0.039* | *0.048* | *0.045* | *0.063* | *0.055* | *0.044* | *0.049* | *0.048* | *0.047* | *0.044* | *0.038* | *0.046* | *0.042* | *0.044* | *0.105* | *0.049* |
| *P-value* | *<.001* | *<.001* | *<.001* | *0.001* | *<.001* | *0.641* | *<.001* | *<.001* | *0.004* | *0.006* | *<.001* | *<.001* | *<.001* | *<.001* | *<.001* | *<.001* | *<.001* |
| Pre-planned orthogonal contrast Probabilities | | | | | | | | | | | | | | | | | |
| Diet 3 *vs.* 4 | 0.687 | 0.366 | 0.971 | 0.999 | 0.476 | 0.816 | 0.915 | 0.741 | 0.861 | 0.888 | 0.499 | 0.573 | 0.518 | 0.447 | 0.906 | 0.971 | 0.744 |
| Diet 5 *vs.* 6 | 0.724 | 0.561 | 0.666 | 0.773 | 0.854 | 0.934 | 0.821 | 0.827 | 0.801 | 0.807 | 0.967 | 0.481 | 0.781 | 0.392 | 0.656 | 0.591 | 0.253 |
| Diet 7 *vs.* 8 | 0.788 | 0.865 | 0.808 | 0.781 | 0.995 | 0.945 | 0.774 | 0.785 | 0.774 | 0.768 | 0.882 | 0.850 | 0.790 | 0.827 | 0.667 | 0.940 | 0.869 |

[a-d] Means within columns not sharing a common suffix are significantly different at P ≤ 0.05.

[1] SBM LCP: low crude protein soybean meal; HCP: high crude protein; CCM: cold-pressed canola meal; ECM: expeller-pressed canola meal.

**Table 8. The influence of dietary treatments[1] on disappearance rate (g/bird/day) of amino acids by the distal jejunum of broiler chickens at 27 days post-hatch.**

| Ingredients | Essential amino acids | | | | | | | | | Non-essential amino acids | | | | | | | Average[2] |
|---|---|---|---|---|---|---|---|---|---|---|---|---|---|---|---|---|---|
| | Arg | His | Ile | Leu | Lys | Met | Phe | Thr | Val | Ala | Asp | Glu | Gly | Pro | Ser | Tyr | |
| Blood meal (1) | 0.463c | 0.649a | 0.479 | 1.422a | 0.633ab | 0.177a | 0.790a | 0.642a | 0.792a | 0.976a | 1.166ab | 1.135b | 0.465 | 0.483abc | 0.583ab | -0.024ab | 10.832 |
| CCM (3) | 0.810b | 0.382b | 0.379 | 0.806b | 0.371bc | 0.198a | 0.451bc | 0.407ab | 0.503b | 0.455b | 0.716bc | 2.401a | 0.538 | 0.676abc | 0.395b | -0.193b | 9.297 |
| ECM (4) | 0.759bc | 0.319b | 0.392 | 0.842b | 0.221c | 0.200a | 0.458bc | 0.376b | 0.496b | 0.456b | 0.617c | 2.321a | 0.475 | 0.583abc | 0.395b | -0.191b | 8.719 |
| Lupins (5) | 1.280a | 0.340b | 0.437 | 0.805b | 0.458bc | 0.052b | 0.446c | 0.373b | 0.399b | 0.347b | 1.120ab | 2.673a | 0.487 | 0.467bc | 0.575ab | 0.231a | 10.491 |
| Peas (6) | 1.325a | 0.299b | 0.449 | 0.848b | 0.932a | 0.064b | 0.548abc | 0.370b | 0.484b | 0.461b | 1.430a | 2.331a | 0.472 | 0.447c | 0.558ab | 0.130a | 11.148 |
| SBM HCP (7) | 0.790bc | 0.381b | 0.585 | 1.120ab | 0.340bc | 0.157a | 0.708ab | 0.477ab | 0.584ab | 0.550b | 1.499a | 2.812a | 0.547 | 0.715a | 0.795a | 0.019ab | 12.079 |
| SBM LCP (8) | 0.885b | 0.380b | 0.553 | 1.066ab | 0.349bc | 0.160a | 0.670abc | 0.444ab | 0.535ab | 0.503b | 1.485a | 2.847a | 0.521 | 0.698ab | 0.719a | 0.027ab | 11.841 |
| SEM | 0.076 | 0.033 | 0.053 | 0.101 | 0.076 | 0.015 | 0.058 | 0.054 | 0.064 | 0.058 | 0.120 | 0.178 | 0.053 | 0.055 | 0.058 | 0.068 | 1.102 |
| P-value | < .001 | < .001 | 0.083 | 0.006 | < .001 | < .001 | 0.003 | 0.013 | 0.006 | < .001 | < .001 | < .001 | 0.872 | 0.002 | 0.001 | 0.006 | 0.293 |
| Pre-planned orthogonal contrast Probabilities | | | | | | | | | | | | | | | | | |
| Diet 3 *vs.* 4 | 0.832 | 0.344 | 0.799 | 0.760 | 0.408 | 0.984 | 0.934 | 0.880 | 0.935 | 0.903 | 0.562 | 0.916 | 0.653 | 0.477 | 0.897 | 0.860 | 0.878 |
| Diet 5 *vs.* 6 | 0.211 | 0.825 | 0.639 | 0.513 | 0.001 | 0.890 | 0.228 | 0.795 | 0.158 | 0.315 | 0.076 | 0.978 | 0.360 | 0.555 | 0.250 | 0.260 | 0.588 |
| Diet 7 *vs.* 8 | 0.402 | 0.686 | 0.758 | 0.683 | 0.864 | 0.984 | 0.644 | 0.826 | 0.370 | 0.658 | 0.935 | 0.673 | 0.421 | 0.736 | 0.336 | 0.437 | 0.730 |

a-c Means within columns not sharing a common suffix are significantly different at P ≤ 0.05

[1]SBM LCP: low crude protein soybean meal; HCP: high crude protein; CCM: cold-pressed canola meal; ECM: expeller-pressed canola meal

[2]Average: the sum of individual amino acids

(P > 0.05). However, there were significant differences between the two canola meal sources for the potential digestible Ile, Leu, Phe, Thr, Val, Ala, Asp and Ser (P < 0.05), with ECM having higher potential digestible of the aforementioned amino acids and total amino acids than CCM.

**Table 9. The influence of dietary treatments[1] on apparent digestibility of amino acids in the proximal ileum in broiler chickens at 27 days post-hatch.**

| Ingredients | Essential amino acids | | | | | | | | | Non-essential amino acids | | | | | | | Average |
|---|---|---|---|---|---|---|---|---|---|---|---|---|---|---|---|---|---|
| | Arg | His | Ile | Leu | Lys | Met | Phe | Thr | Val | Ala | Asp | Glu | Gly | Pro | Ser | Tyr | |
| Blood meal (1) | 0.662b | 0.712bc | 0.695abc | 0.697bc | 0.666bc | 0.766 | 0.688bc | 0.699ab | 0.665abc | 0.711ab | 0.713ab | 0.675b | 0.666bc | 0.661bc | 0.746ab | 0.287cd | 0.669bc |
| CCM (3) | 0.736b | 0.737bc | 0.554d | 0.620c | 0.545cd | 0.765 | 0.632c | 0.530c | 0.562c | 0.614b | 0.600bc | 0.741b | 0.623c | 0.607c | 0.537d | 0.055d | 0.591c |
| ECM (4) | 0.733b | 0.704c | 0.598cd | 0.658c | 0.474d | 0.831 | 0.657c | 0.536c | 0.599c | 0.644b | 0.550c | 0.725b | 0.594c | 0.575c | 0.571cd | 0.063d | 0.595c |
| Lupins (5) | 0.933a | 0.876a | 0.816a | 0.832a | 0.810a | 0.720 | 0.842a | 0.746a | 0.765a | 0.777a | 0.842a | 0.904a | 0.805a | 0.796a | 0.828a | 0.708a | 0.813a |
| Peas (6) | 0.900a | 0.812ab | 0.768ab | 0.796ab | 0.872a | 0.720 | 0.808ab | 0.721ab | 0.753ac | 0.780a | 0.831a | 0.871a | 0.754ab | 0.750ab | 0.779ab | 0.594ab | 0.782ab |
| SBM HCP (7) | 0.706b | 0.721bc | 0.645bcd | 0.670c | 0.546cd | 0.742 | 0.677c | 0.599bc | 0.627bc | 0.656b | 0.657bc | 0.717b | 0.616c | 0.659bc | 0.700b | 0.342bc | 0.642c |
| SBM LCP (8) | 0.714b | 0.726bc | 0.632cd | 0.656c | 0.555cd | 0.722 | 0.668c | 0.585bc | 0.607c | 0.634b | 0.675bc | 0.725b | 0.615c | 0.664bc | 0.674bc | 0.367bc | 0.639c |
| SEM | 0.026 | 0.022 | 0.029 | 0.028 | 0.037 | 0.035 | 0.029 | 0.032 | 0.031 | 0.028 | 0.030 | 0.025 | 0.028 | 0.028 | 0.025 | 0.063 | 0.030 |
| P-value | < .001 | < .001 | < .001 | < .001 | < .001 | 0.293 | < .001 | < .001 | 0.001 | 0.003 | < .001 | < .001 | < .001 | < .001 | < .001 | < .001 | < .001 |
| Pre-planned orthogonal contrast Probabilities | | | | | | | | | | | | | | | | | |
| Diet 3 *vs.* 4 | 0.937 | 0.320 | 0.281 | 0.352 | 0.192 | 0.200 | 0.541 | 0.898 | 0.400 | 0.467 | 0.246 | 0.608 | 0.473 | 0.423 | 0.360 | 0.924 | 0.937 |
| Diet 5 *vs.* 6 | 0.389 | 0.053 | 0.251 | 0.377 | 0.254 | 0.993 | 0.412 | 0.593 | 0.786 | 0.951 | 0.793 | 0.368 | 0.218 | 0.262 | 0.187 | 0.211 | 0.475 |
| Diet 7 *vs.* 8 | 0.822 | 0.878 | 0.753 | 0.729 | 0.859 | 0.691 | 0.829 | 0.772 | 0.642 | 0.596 | 0.687 | 0.831 | 0.985 | 0.906 | 0.471 | 0.786 | 0.929 |

a-d Means within columns not sharing a common suffix are significantly different at P ≤ 0.05.

[1]SBM LCP: low crude protein soybean meal; HCP: high crude protein; CCM: cold-pressed canola meal; ECM: expeller-pressed canola meal.

**Table 10. The influence of dietary treatments[1] on disappearance rate (g/bird/day) of amino acids by the proximal ileum in broiler chickens at 27 days post-hatch.**

| Ingredients | Essential amino acids | | | | | | | | | | Non-essential amino acids | | | | | | Average[2] |
|---|---|---|---|---|---|---|---|---|---|---|---|---|---|---|---|---|---|
| | Arg | His | Ile | Leu | Lys | Met | Phe | Thr | Val | Ala | Asp | Glu | Gly | Pro | Ser | Tyr | |
| Blood meal (1) | 0.605[c] | 0.751[a] | 0.585[b] | 1.673[a] | 0.827[ab] | 0.223[ab] | 0.939[a] | 0.769[a] | 0.951[a] | 1.146[a] | 1.419[cd] | 1.427[d] | 0.553[bc] | 0.574[b] | 0.712[b] | 0.170[ab] | 13.32[abc] |
| CCM (3) | 0.997[b] | 0.479[bc] | 0.514[b] | 1.055[c] | 0.590[bc] | 0.224[ab] | 0.587[b] | 0.554[bcd] | 0.667[bc] | 0.612[bc] | 1.018[de] | 2.993[bc] | 0.737[a] | 0.859[a] | 0.557[b] | 0.026[b] | 12.47[c] |
| ECM (4) | 0.970[b] | 0.450[bc] | 0.583[b] | 1.173[bc] | 0.445[c] | 0.259[a] | 0.636[b] | 0.587[bc] | 0.745[bc] | 0.672[bc] | 0.962[e] | 3.008[abc] | 0.742[a] | 0.849[a] | 0.616[b] | 0.033[b] | 12.79[bc] |
| Lupins (5) | 1.354[a] | 0.369[cd] | 0.496[b] | 0.895[c] | 0.526[c] | 0.065[b] | 0.502[b] | 0.411[d] | 0.452[d] | 0.403[d] | 1.247[cde] | 2.855[bc] | 0.531[c] | 0.500[b] | 0.633[b] | 0.300[a] | 11.54[c] |
| Peas (6) | 1.401[a] | 0.332[c] | 0.511[b] | 0.955[c] | 1.021[a] | 0.080[b] | 0.614[b] | 0.422[cd] | 0.554[cd] | 0.523[cd] | 1.582[bc] | 2.526[c] | 0.529[c] | 0.499[bb] | 0.629[b] | 0.220[a] | 12.40[c] |
| SBM HCP (7) | 1.052[b] | 0.485[b] | 0.791[a] | 1.462[ab] | 0.567[c] | 0.206[b] | 0.915[a] | 0.643[ab] | 0.800[ab] | 0.744[b] | 1.964[a] | 3.468[ab] | 0.707[ab] | 0.900[a] | 1.032[a] | 0.247[a] | 15.98[ab] |
| SBM LCP (8) | 1.152[ab] | 0.505[b] | 0.786[a] | 1.458[ab] | 0.595[bc] | 0.210[b] | 0.914[a] | 0.641[ab] | 0.779[ab] | 0.720[b] | 2.073[a] | 3.656[a] | 0.723[a] | 0.931[a] | 0.987[a] | 0.275[a] | 16.41[a] |
| SEM | 0.062 | 0.025 | 0.036 | 0.073 | 0.056 | 0.009 | 0.043 | 0.038 | 0.045 | 0.043 | 0.091 | 0.139 | 0.037 | 0.040 | 0.043 | 0.041 | 0.793 |
| P-value | <.001 | <.001 | <.001 | <.001 | <.001 | <.001 | <.001 | <.001 | <.001 | <.001 | <.001 | <.001 | 0.004 | <.001 | <.001 | <.001 | 0.002 |
| Pre-planned orthogonal contrast Probabilities | | | | | | | | | | | | | | | | | |
| Diet 3 vs. 4 | 0.811 | 0.382 | 0.321 | 0.418 | 0.162 | 0.114 | 0.431 | 0.679 | 0.336 | 0.501 | 0.514 | 0.943 | 0.947 | 0.842 | 0.450 | 0.917 | 0.929 |
| Diet 5 vs. 6 | 0.696 | 0.125 | 0.723 | 0.756 | 0.007 | 0.275 | 0.074 | 0.623 | 0.122 | 0.611 | 0.046 | 0.037 | 0.365 | 0.335 | 0.704 | 0.713 | 0.977 |
| Diet 7 vs. 8 | 0.862 | 0.321 | 0.466 | 0.509 | 0.450 | 0.809 | 0.991 | 0.539 | 0.752 | 0.889 | 0.284 | 0.156 | 0.457 | 0.163 | 0.746 | 0.668 | 0.478 |

[a-d] Means within columns not sharing a common suffix are significantly different at P ≤ 0.05.

[1] SBM LCP: low crude protein soybean meal; HCP: high crude protein; CCM: cold-pressed canola meal; ECM: expeller-pressed canola meal.

[2] Average: the sum of individual amino acids.

**Table 11. The influence of dietary treatments[1] on apparent digestibility of amino acids in distal ileum in broiler chickens at 27 days post-hatch.**

| Ingredients | Essential amino acids | | | | | | | | | | Non-essential amino acids | | | | | | |
|---|---|---|---|---|---|---|---|---|---|---|---|---|---|---|---|---|---|
| | Arg | His | Ile | Leu | Lys | Met | Phe | Thr | Val | Ala | Asp | Glu | Gly | Pro | Ser | Tyr | Average |
| Blood meal (1) | 0.773[b] | 0.780[b] | 0.769[ab] | 0.777[ab] | 0.767[bc] | 0.848[ab] | 0.768[bc] | 0.789[a] | 0.747[abc] | 0.787 | 0.796[ab] | 0.762[c] | 0.746[ab] | 0.741[ab] | 0.812[a] | 0.503[bc] | 0.760[abc] |
| CCM (3) | 0.814[b] | 0.787[b] | 0.670[b] | 0.734[b] | 0.680[cd] | 0.862[ab] | 0.741[c] | 0.624[b] | 0.657[c] | 0.720 | 0.702[cd] | 0.808[bc] | 0.708[b] | 0.666[bc] | 0.648[b] | 0.336[cd] | 0.697[c] |
| ECM (4) | 0.818[b] | 0.756[b] | 0.705[bc] | 0.763[b] | 0.563[d] | 0.898[a] | 0.775[abc] | 0.606[b] | 0.679[bc] | 0.731 | 0.670[d] | 0.806[bc] | 0.672[b] | 0.621[c] | 0.667[b] | 0.222[d] | 0.685[c] |
| Lupins (5) | 0.945[a] | 0.876[a] | 0.847[a] | 0.860[a] | 0.848[ab] | 0.765[b] | 0.865[a] | 0.785[a] | 0.803[a] | 0.794 | 0.864[a] | 0.920[a] | 0.833[a] | 0.832[a] | 0.848[a] | 0.767[a] | 0.841[a] |
| Peas (6) | 0.917[a] | 0.825[ab] | 0.794[ab] | 0.822[ab] | 0.897[a] | 0.774[b] | 0.834[ab] | 0.746[a] | 0.774[ab] | 0.801 | 0.850[a] | 0.892[ab] | 0.774[ab] | 0.772[a] | 0.801[a] | 0.676[ab] | 0.809[ac] |
| SBM HCP (7) | 0.798[b] | 0.784[b] | 0.750[abc] | 0.767[b] | 0.672[cd] | 0.826[ab] | 0.777[abc] | 0.684[ab] | 0.730[abc] | 0.743 | 0.737[bcd] | 0.793[c] | 0.697[b] | 0.734[ab] | 0.782[a] | 0.526[bc] | 0.738[bc] |
| SBM LCP (8) | 0.809[b] | 0.794[ab] | 0.751[abc] | 0.764[b] | 0.695[c] | 0.826[ab] | 0.779[abc] | 0.685[ab] | 0.723[abc] | 0.732 | 0.767[bc] | 0.812[bc] | 0.711[b] | 0.754[ab] | 0.772[a] | 0.566[ab] | 0.747[abc] |
| *SEM* | *0.019* | *0.018* | *0.022* | *0.021* | *0.028* | *0.024* | *0.021* | *0.024* | *0.024* | *0.023* | *0.018* | *0.019* | *0.023* | *0.022* | *0.021* | *0.049* | *0.023* |
| P-value | <.001 | 0.001 | <.001 | 0.004 | <.001 | 0.005 | 0.002 | <.001 | 0.001 | 0.075 | <.001 | <.001 | 0.005 | <.001 | <.001 | <.001 | 0.003 |
| **Pre-planned orthogonal contrast Probabilities** | | | | | | | | | | | | | | | | | |
| Diet 3 vs. 4 | 0.872 | 0.238 | 0.265 | 0.336 | 0.006 | 0.309 | 0.255 | 0.593 | 0.515 | 0.750 | 0.231 | 0.963 | 0.290 | 0.168 | 0.516 | 0.112 | 0.700 |
| Diet 5 vs. 6 | 0.323 | 0.062 | 0.099 | 0.224 | 0.231 | 0.807 | 0.293 | 0.277 | 0.407 | 0.825 | 0.595 | 0.310 | 0.086 | 0.070 | 0.126 | 0.202 | 0.347 |
| Diet 7 vs. 8 | 0.676 | 0.682 | 0.971 | 0.910 | 0.572 | 0.987 | 0.940 | 0.965 | 0.834 | 0.750 | 0.253 | 0.492 | 0.669 | 0.541 | 0.746 | 0.567 | 0.787 |

[a-d] Means within columns not sharing a common suffix are significantly different at P ≤ 0.05.

[1] SBM LCP: low crude protein soybean meal; HCP: high crude protein; CCM: cold-pressed canola meal; ECM: expeller-pressed canola meal.

**Table 12. The influence of dietary treatments[1] on disappearance rate (g/bird/day) of amino acids by distal ileum in broiler chickens at 27 days post-hatch.**

| Ingredients | Essential amino acids | | | | | | | | | Non-essential amino acids | | | | | | | Average[2] |
|---|---|---|---|---|---|---|---|---|---|---|---|---|---|---|---|---|---|
| | Arg | His | Ile | Leu | Lys | Met | Phe | Thr | Val | Ala | Asp | Glu | Gly | Pro | Ser | Tyr | |
| Blood meal (1) | 0.706$^c$ | 0.823$^a$ | 0.647$^{bc}$ | 1.866$^a$ | 0.952$^{ab}$ | 0.247$^{ab}$ | 1.049 | 0.867$^a$ | 1.069$^a$ | 1.270$^a$ | 1.584$^b$ | 1.611$^e$ | 0.620$^b$ | 0.645$^b$ | 0.776$^b$ | 0.295$^{abc}$ | 15.03$^{bc}$ |
| CCM (3) | 1.102$^b$ | 0.512$^b$ | 0.622$^{bc}$ | 1.249$^{bc}$ | 0.738$^{bc}$ | 0.252$^{ab}$ | 0.688 | 0.653$^b$ | 0.779$^b$ | 0.718$^b$ | 1.191$^c$ | 3.262$^{bc}$ | 0.837$^a$ | 0.942$^a$ | 0.721$^b$ | 0.174$^{cd}$ | 14.39$^c$ |
| ECM (4) | 1.100$^b$ | 0.483$^b$ | 0.690$^b$ | 1.351$^b$ | 0.530$^c$ | 0.281$^a$ | 0.746 | 0.655$^b$ | 0.845$^b$ | 0.756$^b$ | 1.166$^c$ | 3.371$^{bc}$ | 0.829$^a$ | 0.916$^a$ | 0.672$^b$ | 0.118$^d$ | 14.66$^c$ |
| Lupins (5) | 1.371$^a$ | 0.368$^c$ | 0.514$^c$ | 0.923$^d$ | 0.550$^c$ | 0.069$^c$ | 0.515 | 0.432$^c$ | 0.473$^c$ | 0.411$^c$ | 1.278$^{bc}$ | 2.904$^{cd}$ | 0.548$^c$ | 0.520$^b$ | 0.648$^b$ | 0.323$^{ab}$ | 11.85$^c$ |
| Peas (6) | 1.428$^a$ | 0.338$^c$ | 0.528$^c$ | 0.985$^{cd}$ | 1.050$^a$ | 0.087$^c$ | 0.634 | 0.436$^c$ | 0.568$^c$ | 0.537$^c$ | 1.619$^b$ | 2.585$^d$ | 0.543$^c$ | 0.512$^b$ | 0.646$^b$ | 0.249$^{bcd}$ | 12.74$^c$ |
| SBM HCP (7) | 1.190$^{ab}$ | 0.528$^b$ | 0.921$^a$ | 1.676$^a$ | 0.701$^c$ | 0.230$^b$ | 1.050 | 0.736$^{ab}$ | 0.932$^{ab}$ | 0.844$^b$ | 2.203$^a$ | 3.837$^{ab}$ | 0.801$^a$ | 1.004$^a$ | 1.153$^a$ | 0.379$^{ab}$ | 18.18$^{ab}$ |
| SBM LCP (8) | 1.308$^{ab}$ | 0.554$^b$ | 0.939$^a$ | 1.703$^a$ | 0.750$^{bc}$ | 0.241$^b$ | 1.069 | 0.753$^{ab}$ | 0.933$^{ab}$ | 0.835$^b$ | 2.365$^a$ | 4.106$^a$ | 0.840$^a$ | 1.062$^a$ | 1.134$^a$ | 0.429$^a$ | 19.02$^a$ |
| *SEM* | *0.060* | *0.025* | *0.033* | *0.067* | *0.051* | *0.007* | *0.039* | *0.034* | *0.041* | *0.042* | *0.083* | *0.135* | *0.035* | *0.039* | *0.038* | *0.034* | *0.732* |
| P-value | < .001 | < .001 | < .001 | < .001 | < .001 | < .001 | < .001 | < .001 | < .001 | < .001 | < .001 | < .001 | < .001 | < .001 | < .001 | < .001 | < .001 |
| Pre-planned orthogonal contrast Probabilities | | | | | | | | | | | | | | | | | |
| Diet 3 *vs.* 4 | 0.866 | 0.372 | 0.226 | 0.318 | 0.006 | 0.119 | 0.339 | 0.969 | 0.361 | 0.539 | 0.835 | 0.586 | 0.875 | 0.515 | 0.469 | 0.188 | 0.935 |
| Diet 5 *vs.* 6 | 0.502 | 0.399 | 0.763 | 0.524 | 0.001 | 0.112 | 0.082 | 0.934 | 0.117 | 0.035 | 0.006 | 0.100 | 0.918 | 0.879 | 0.974 | 0.113 | 0.391 |
| Diet 7 *vs.* 8 | 0.172 | 0.478 | 0.703 | 0.774 | 0.498 | 0.335 | 0.740 | 0.717 | 0.992 | 0.876 | 0.171 | 0.163 | 0.443 | 0.285 | 0.722 | 0.282 | 0.425 |

[a-d] Means within columns not sharing a common suffix are significantly different at P ≤ 0.05.

[1] SBM LCP: low crude protein soybean meal; HCP: high crude protein; CCM: cold-pressed canola meal; ECM: expeller-pressed canola meal.

[2] Average: the sum of individual amino acids.

## Discussion

According to Liu and Selle [6], balanced availabilities of glucose and amino acids at the sites of protein synthesis is important for optimal feed conversion efficiency; therefore, not only the extent but also the rate and site of protein and amino acid digestion should be considered. The present study is one of the series [1–3] determining digestion rates of protein and amino acids

**Table 13. The influence of dietary treatments[1] on potential digestible amino acids (g/g) in broiler chickens at 27 days post-hatch.**

| Ingredients | Essential amino acids | | | | | | | | | Non-essential amino acids | | | | | | | Average |
|---|---|---|---|---|---|---|---|---|---|---|---|---|---|---|---|---|---|
| | Arg | His | Ile | Leu | Lys | Met | Phe | Thr | Val | Ala | Asp | Glu | Gly | Pro | Ser | Tyr | |
| Blood meal (1) | 0.780 | 0.795 | 0.788 | 0.795 | 0.775 | 0.868 | 0.786 | 0.797 | 0.768 | 0.806 | 0.806 | 0.777 | 0.764 | 0.760 | 0.820 | 0.783 | 0.773 |
| CCM (3) | 0.834 | 0.815 | 0.779 | 0.793 | 0.962 | 0.860 | 0.790 | 0.715 | 0.742 | 0.794 | 0.790 | 0.826 | 0.785 | 0.719 | 0.767 | 1.000 | 0.807 |
| ECM (4) | 0.936 | 0.896 | 0.967 | 0.946 | 0.982 | 0.908 | 0.966 | 0.914 | 0.942 | 0.949 | 0.973 | 0.914 | 0.937 | 0.852 | 0.963 | 0.982 | 0.961 |
| Lupins (5) | 0.938 | 0.888 | 0.855 | 0.872 | 0.871 | 0.973 | 0.877 | 0.812 | 0.836 | 0.840 | 0.864 | 0.917 | 0.845 | 0.839 | 0.854 | 0.879 | 0.857 |
| Peas (6) | 0.888 | 0.810 | 0.801 | 0.816 | 0.852 | 0.977 | 0.819 | 0.767 | 0.805 | 0.801 | 0.816 | 0.860 | 0.769 | 0.765 | 0.790 | 0.986 | 0.810 |
| SBM HCP (7) | 0.816 | 0.809 | 0.849 | 0.858 | 0.892 | 0.891 | 0.861 | 0.742 | 0.843 | 0.775 | 0.775 | 0.813 | 0.752 | 0.772 | 0.798 | 0.892 | 0.850 |
| SBM LCP (8) | 0.886 | 0.867 | 0.860 | 0.863 | 0.877 | 0.912 | 0.868 | 0.834 | 0.850 | 0.855 | 0.863 | 0.883 | 0.843 | 0.859 | 0.864 | 0.854 | 0.861 |
| SEM | 0.046 | 0.042 | 0.051 | 0.048 | 0.053 | 0.038 | 0.046 | 0.062 | 0.056 | 0.052 | 0.049 | 0.047 | 0.057 | 0.059 | 0.050 | 0.087 | 0.051 |
| P-value | 0.153 | 0.462 | 0.171 | 0.303 | 0.145 | 0.236 | 0.123 | 0.365 | 0.262 | 0.282 | 0.107 | 0.305 | 0.253 | 0.544 | 0.138 | 0.526 | 0.239 |
| Pre-planned orthogonal contrast Probabilities | | | | | | | | | | | | | | | | | |
| Diet 3 *vs.* 4 | 0.124 | 0.181 | 0.012 | 0.034 | 0.793 | 0.367 | 0.010 | 0.029 | 0.016 | 0.040 | 0.011 | 0.190 | 0.064 | 0.118 | 0.008 | 0.885 | 0.038 |
| Diet 5 *vs.* 6 | 0.467 | 0.219 | 0.475 | 0.436 | 0.807 | 0.933 | 0.399 | 0.624 | 0.710 | 0.607 | 0.508 | 0.416 | 0.365 | 0.398 | 0.391 | 0.403 | 0.539 |
| Diet 7 *vs.* 8 | 0.282 | 0.339 | 0.884 | 0.937 | 0.843 | 0.694 | 0.910 | 0.298 | 0.925 | 0.277 | 0.207 | 0.296 | 0.261 | 0.302 | 0.357 | 0.755 | 0.876 |

[1] SBM LCP: low crude protein soybean meal; HCP: high crude protein; CCM: cold-pressed canola meal; ECM: expeller-pressed canola meal.

in common feed ingredients used in Australia to explore the possibility of considering digestion rate in practical feed formulation. The protein and amino acid digestion rates were predicted by fitting exponential models to describe the relationship between apparent digestibility coefficients and their corresponding mean retention time in various segments of the small intestine. The digestibility coefficient of nitrogen and most of the amino acids in all four sections of the small intestine were different across the diets with larger variations detected in proximal jejunum. Nitrogen and amino acid disappearance rates also followed similar pattern and the values calculated significantly differed across diets. The digestibility coefficients of crude protein and amino acids determined in terminal ileum were very close and slightly higher than the ileal digestibility coefficients reported in the literature for all the ingredients [20]. The highest predicted protein digestion rate was measured in broiler chickens offered the BM diet followed by SBM HCP and lupins. The pattern of nitrogen disappearance rate measured in four sections of small intestine was quite similar to the average amino acid disappearance rate. However, disappearance rates were lower than their corresponding nitrogen disappearance rates in the proximal and distal jejunum. The magnitude of differences detected became smaller in the proximal and distal ileum.

CCM had higher Lys digestibility and disappearance rate in the distal ileum than ECM. In addition, Lys digestion rate in CCM was numerically higher than ECM by nearly two fold. These differences between the two meal samples highlights the impact various processing conditions, particularly high temperature, could have on Lys utilisation. Classen et al. [21] reported that Maillard reactions usually occur in canola meal during desolventization and toasting when the meal temperature and moisture content are at least 105˚C and 10%, respectively. Although, ECM is not subjected to a desolventizing/toasting steps as is solvent extracted meal; but expeller meal is still subjected to the potential effects of pre-press seed heating and heat due to the friction generated during the expelling process, favoring the conditions for secondary and tertiary Maillard reactions to occur. Conversely, there is no in-put heat during cold-press extraction, leaving the Lys content better preserved from heat damage [22].

There were differences between peas and lupins in apparent disappearance rates of Lys in the four small intestinal segments, Ala and Glu in the proximal ileum and Asp and Ala in the distal ileum. This could largely be attributed to the inherent properties and the presence of different anti-nutritional factors in these legume seeds. Lupins contain more crude fat and fiber, and less starch than peas, and also their non-starch polysaccharides profile is different [23]. All these differences may have contributed to the superior amino acid digestibilities observed in peas compared to lupins [24]. Moreover, the calculated fibre content of lupins diet was higher than the pea diet by almost two fold (107 vs 51.1 g/kg). This high dietary fibre could have influenced the digesta passage rate, retention time and eventually amino acid disappearance rates. As there were significant negative correlations between calculated diet crude fibre content with the average amino acid disappearance (r = -0.54, P < 0.05) and digestion rates (r = -0.29, P < 0.05) in the distal ileum. The negative correlations between calculated dietary fibre and protein disappearance rates at the proximal jejunum (r = -0.40) and proximal ileum (r = -0.39) may also explain the protein disappearance rate differences between SBM HCP and SBM LCP, as the latter diet fibre content was nearly three times higher than the SBM HCP diet.

The digestive dynamics of protein-bound and non-bound (synthetic or crystalline) amino acids are substantially different, with the latter being absorbed more rapidly [11, 25]. Therefore, to evade any confounding effect of non-bound amino acids on protein and amino acid digestion rates they were not included in the experimental diets. As expected, the experimental diets compromised the growth performance of broiler chickens. Growth performance results are usually not reported in digestibility studies [26–28]; however, they are included herein to provide complete information on potential confounding factors in relation to digestibility

results determined in the present study. It is important to note that during the 21–28 days post-hatch experimental period, that birds offered experimental diets did not lose body weight as the atypical experimental diets with the test ingredient as the sole protein source may cause body weight loss and influence the accuracy of the data. The amino acid profile of each protein meal was reflected in the test diets; thus, diets based on protein meals, which were better balanced such as CM and SBM, generated higher body weight over the 7-day growth period. Protein meals of animal origin in the present study have higher levels of essential amino acids such as leucine and phenylalanine (Table 1), resulting in an imbalanced amino acids profile particularly when included at high levels. This disparity may also explain the lower body weight gain of birds offered BM and PPM diets in comparison to chickens offered SBM and CM diets. The inferior performance of birds on PPM diet *vs.* BM diets is most likely due to the negative effect of excess dietary sodium in PPM diet on feed intake (1.6 *vs.* 23.7 g/kg). High dietary sodium has been shown to linearly decrease feed intake and body weight gain in broiler chickens [29].

The experimental diets used in this study were formulated to be iso-caloric except for lupins which had approximately 0.3 MJ/kg higher AME than the other diets. This higher formulated AME was also reflected on determined AME values, although this was not necessarily accompanied by a higher AME:GE ratio for lupins. It has been suggested that the ratio between AME and gross energy is more indicative for energy efficiency [30]. Birds offered the PPM diet recorded both a high AME and the highest AME: GE ratio. This discrepancy in determined AME was mostly likely to do with the variation in feed intake. Dietary analysed crude protein was negatively correlated with the determined AME (r = -0.41, P = 0.012) and AMEn (r = -0.58, P < 0.001). Similarly, Jackson et al. [31] reported that increasing dietary protein ingestion depressed protein and energy utilisation, and suggested that protein and energy utilisation were negatively correlated with protein intake. Moreover, the variation in soybean oil inclusions in the experimental diets in order to make the diets iso-energetic may have also confounded the energy utilisation and digestive dynamics results. The possibility of lipid influence digestive dynamics could be due to its impact of feed intake and gastric emptying. Liu et al. [32] reported that broiler chickens offered diets containing 8.5% higher lipid concentrations generated an 8.8% reduction in feed intake. Similar to the 'ileal brake' mechanism described in mammals, Martinez et al. [33] reported that intraluminal infusion of lipids in poultry modulates gastrointestinal motility including an increase in duodenogastric refluxes or episodes of reverse peristalsis and these actions could delay gastric emptying and increase transit time. Overall, both growth performance and parameters of energy utilizations are imperfect indicators of the quality of diet protein sources because they were most certainly confounded by the nature of the experimental diets.

## Conclusions

In conclusion, the data presented in this study demonstrated that the predicted protein and amino acid disappearance rates and digestibilities along the small intestine vary among different protein-rich ingredients. This variation was smaller between similar ingredients processed with differing methods than from different sources. The present study detected more variations in jejunal amino acid and protein digestibilities in comparison to ileal digestibilities. This emphasises that both the extent and rate of protein and amino acid digestion must be considered as indicators of protein quality. This initial assessment also provides valuable information for future research in digestive dynamics and growth performance in poultry, where experimental diets will be formulated with the same quantity of digestible protein or amino acids but with different rates of protein and amino acid digestions.

## Supporting information

**S1 Data.**
(XLSX)

## Acknowledgments

The authors also would like to thank Ms Joy Gill, Ms Kylie Warr and Mr Duwei Chen within the Poultry Research Foundation for their technical support.

## Author Contributions

**Conceptualization:** P. H. Selle, S. Y. Liu.

**Formal analysis:** M. Toghyani, L. R. McQuade, B. V. Mclnerney.

**Funding acquisition:** S. Y. Liu.

**Investigation:** A. F. Moss, S. Y. Liu.

**Methodology:** L. R. McQuade, B. V. Mclnerney, A. F. Moss.

**Project administration:** S. Y. Liu.

**Writing – original draft:** M. Toghyani.

**Writing – review & editing:** L. R. McQuade, B. V. Mclnerney, A. F. Moss, P. H. Selle, S. Y. Liu.

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
