## [Decision Letter · Decision Letter 0]

17 Jul 2020

PONE-D-20-12708

Initial assessment of protein and amino acid digestive dynamics in protein-rich feedstuffs for broiler chickens

PLOS ONE

Dear Dr. Liu,

Thank you for submitting your manuscript to PLOS ONE. After careful consideration, we feel that it has merit but does not fully meet PLOS ONE’s publication criteria as it currently stands. Therefore, we invite you to submit a revised version of the manuscript that addresses the points raised during the review process.

We look forward to receiving your revised manuscript.

Kind regards,

Juan J Loor

Academic Editor

PLOS ONE

Journal Requirements:

Reviewers' comments:

Reviewer's Responses to Questions

**Comments to the Author**

1. Is the manuscript technically sound, and do the data support the conclusions?

Reviewer #1: Yes

Reviewer #2: Yes

Reviewer #3: Partly

2. Has the statistical analysis been performed appropriately and rigorously? 

Reviewer #1: Yes

Reviewer #2: Yes

Reviewer #3: Yes

3. Have the authors made all data underlying the findings in their manuscript fully available?

Reviewer #1: Yes

Reviewer #2: Yes

Reviewer #3: Yes

4. Is the manuscript presented in an intelligible fashion and written in standard English?

Reviewer #1: Yes

Reviewer #2: Yes

Reviewer #3: Yes

5. Review Comments to the Author

Reviewer #1: In this manuscript, Toghyani and colleagues evaluated protein and amino acid digestibility in protein-rich feedstuffs in broiler chickens. This is a well studied and written manuscript. The results bring new insights into poultry protein and amino acid nutrition and beyond. The following are minor comments.

1. In table 4, 6, 8, and 10, the disappearance rate data are very interesting and insightful. In table 4, the value of either distal ileum apparent digestibility coefficient (0.714) or disappearance rate (19.60) for blood meal is wrong, and 0.714 seems a little small. Please make a correction and double-check other data.

2. In tables 6, 8, and 10 for AA disappearance rate, how were the values in the “Average” column calculated? It seems that the individual AA values were added together for the “Average”. It would be better to put a notation in the table.

3. Despite variable AA disappearance rate for the protein-rich ingredients across intestinal segments, the protein disappearance rate seems increased from proximal jejunum to distal ileum for the ingredients. My impression is that most nutrients are absorbed in jejuni or upper ileum, and the protein disappearance rate would be decreased from proximal jejunum to distal ileum. Do you have any insight for your observations? Is it possible that microbiota activities play any role?

Reviewer #2: Line 88: Please further describe the cold pelleting process. Specifically, equipment utilized and any parameters measured (e.g. temperature rise due to friction) that could potentially impact protein structure.

Reviewer #3: Initial assessment of protein and amino acid digestive dynamics in protein-rich feedstuffs for broiler chickens

This is a generally well-written paper that seeks to extend the concept of opportunities to synchronize dietary amino acid and energy availability to optimize feed efficiency in broilers. A perfect experiment with no confounding factors would be quite difficult, if not impossible, to design to address this issues. Though several potentially confounding factors were addressed by the authors, a few more need attention in interpretation of the data.

1. The drastically different oil contents of the diets is certainly confounding to the energy utilization, and perhaps protein/AA digestion and absorption, due to well established effects of lipid supplementation on rate of passage and nutrient utilization. In the current experiment, simple linear regression between dietary oil content and AME of the diets yields an R2 of 0.66. Deleting the data from the lupin group, which contained 1.9 to 3.5x more oil than the other diets, the R2 for this relationship is still 0.36. Therefore, it is likely that much of the variation in the energy utilization can be explained by oil content alone, despite being isocaloric. While AME determination was not the primary objective, the potential for this to affect AA utilization cannot be ignored, especially since correlations between AME and CP utilization are addressed (L457)

2. Was the potential for reverse peristalsis considered? It is not mentioned anywhere in the manuscript. Is the calculation of mean retention time based on digesta weight valid when digesta can travel in both directions, which would not be accounted in the current method. This was similarly done by Bryan et al 2019, and they cited Wilson and Leibholz who did work in pigs which do not have the antiperistaltic activity that birds do. Mean retention time estimates reported in the literature based on excreta are obviously not challenged with potential for additional retrograde movement. Can the authors provide justification/validation for this approach elsewhere in the literature?

a. Further, the accuracy of these methods is entirely dependent on similar kinetics of the marker and ingredient-derived AA. This likely evens out a bit by the ileum or excreta, but again, is likely greatly influenced by reverse peristalsis in duodenum, affecting estimates in jejunum.

3. Per previous comment, this paper is very similar to the work conducted by Bryan et al. 2019 so the advancements, similarities, and differences between these 2 works should be addressed beyond the current text beginning in L396.

4. There are likely sampling location and ingredient-dependent effects (e.g., fiber and mucin-associated AA thr, glyc, etc per discussion in L425-429) on endogenous AA losses, which were not accounted for in the experiment or considered in the interpretation. This should be addressed.

a. Were digesta squeezed or flushed to minimize epithelial sloughing/endogenous losses?

5. Given the amount and complexity of the data, there seems to be quite a bit of discussion around secondary points that could be replaced with more directly relevant topics. Examples include L404 (processing effects on Lys dig); L434 to L449 (growth performance); L450 to 462 (energy utilization per comment above).

Additional comments

- It would be helpful to keep the order of treatments in the diet table similar to that presented for the data tables.

- “Potential digestible protein” is not clearly defined – is this the same as the asymptote described in line 189?

- Provide units for Table 4 “disappearance rates”

- Why were plasma data omitted from tables after Table 6?

- Is Table 13 necessary? There are no differences and the SEM is larger than the means reported in many cases, indicated data not incredibly useful.

6. PLOS authors have the option to publish the peer review history of their article (what does this mean?). If published, this will include your full peer review and any attached files.

Reviewer #1: No

Reviewer #2: No

Reviewer #3: No

---

## [Author Response · Author response to Decision Letter 0]

26 Jul 2020

Reviewer #1: In this manuscript, Toghyani and colleagues evaluated protein and amino acid digestibility in protein-rich feedstuffs in broiler chickens. This is a well studied and written manuscript. The results bring new insights into poultry protein and amino acid nutrition and beyond. The following are minor comments.

1. In table 4, 6, 8, and 10, the disappearance rate data are very interesting and insightful. In table 4, the value of either distal ileum apparent digestibility coefficient (0.714) or disappearance rate (19.60) for blood meal is wrong, and 0.714 seems a little small. Please make a correction and double-check other data.

Thank you very much for pointing this out. We have double checked and corrected this. Sorry about this mistake.

2. In tables 6, 8, and 10 for AA disappearance rate, how were the values in the “Average” column calculated? It seems that the individual AA values were added together for the “Average”. It would be better to put a notation in the table.

Included as footnote

3. Despite variable AA disappearance rate for the protein-rich ingredients across intestinal segments, the protein disappearance rate seems increased from proximal jejunum to distal ileum for the ingredients. My impression is that most nutrients are absorbed in jejuni or upper ileum, and the protein disappearance rate would be decreased from proximal jejunum to distal ileum. Do you have any insight for your observations? Is it possible that microbiota activities play any role?

Yes, most glucose and amino acids are absorbed in jejunum. The reason the disappearance rate in ileum is still higher than jejunum is because this is accumulated disappearance rate. More accurately, it is disappearance rate by distal ileum. We have changed this in the Table to be clear.

Reviewer #2: Line 88: Please further describe the cold pelleting process. Specifically, equipment utilized and any parameters measured (e.g. temperature rise due to friction) that could potentially impact protein structure.

Included 

Reviewer #3: Initial assessment of protein and amino acid digestive dynamics in protein-rich feedstuffs for broiler chickens

This is a generally well-written paper that seeks to extend the concept of opportunities to synchronize dietary amino acid and energy availability to optimize feed efficiency in broilers. A perfect experiment with no confounding factors would be quite difficult, if not impossible, to design to address this issues. Though several potentially confounding factors were addressed by the authors, a few more need attention in interpretation of the data.

Thank you.

1. The drastically different oil contents of the diets is certainly confounding to the energy utilization, and perhaps protein/AA digestion and absorption, due to well established effects of lipid supplementation on rate of passage and nutrient utilization. In the current experiment, simple linear regression between dietary oil content and AME of the diets yields an R2 of 0.66. Deleting the data from the lupin group, which contained 1.9 to 3.5x more oil than the other diets, the R2 for this relationship is still 0.36. Therefore, it is likely that much of the variation in the energy utilization can be explained by oil content alone, despite being isocaloric. While AME determination was not the primary objective, the potential for this to affect AA utilization cannot be ignored, especially since correlations between AME and CP utilization are addressed (L457)

Thank you for this suggestion and the below is included,

Moreover, the variation in soybean oil inclusions in the experimental diets in order to make the diets iso-energetic may have also confounded the energy utilisation and digestive dynamics results. The possibility of lipid influence digestive dynamics could be due to its impact of feed intake and gastric emptying. Liu et al. [31] reported that broiler chickens offered diets containing 8.5% higher lipid concentrations generated an 8.8% reduction in feed intake. Similar to the ‘ileal brake’ mechanism described in mammals, Martinez et al. [32] reported that intraluminal infusion of lipids in poultry modulates gastrointestinal motility including an increase in duodenogastric refluxes or episodes of reverse peristalsis and these actions could delay gastric emptying and increase transit time.

2. Was the potential for reverse peristalsis considered? It is not mentioned anywhere in the manuscript. Is the calculation of mean retention time based on digesta weight valid when digesta can travel in both directions, which would not be accounted in the current method. This was similarly done by Bryan et al 2019, and they cited Wilson and Leibholz who did work in pigs which do not have the antiperistaltic activity that birds do. Mean retention time estimates reported in the literature based on excreta are obviously not challenged with potential for additional retrograde movement. Can the authors provide justification/validation for this approach elsewhere in the literature?

Unfortunately, reverse peristalsis is a confounding factor and we hope to minimise by sampling from every birds in one cage/group. Mean retention time determined by appreance of marker in the excreta is certainly the most reliable method. However, this method is not practical to quantify retention time in individual intestinal segments. The current method is originated from the below two references and they are included in M&M. 

17. Weurding RE, Veldman A, Veen WAG, van der Aar PJ, Verstegen MWA. Starch digestion rate in the small intestine of broiler chickens differs among feedstuffs. J Nutr. 2001;131(9):2329-35. PubMed PMID: ISI:000170921000018.

18. Enting H, Pos J, Weurding RE, Veldman A. Starch digestion rate affects broiler performance. Proc Aust Poult Sci Symp. 2005;17:17-20. PubMed PMID: CABI:20073008813; PubMed Central PMCID: PMC2005.

a. Further, the accuracy of these methods is entirely dependent on similar kinetics of the marker and ingredient-derived AA. This likely evens out a bit by the ileum or excreta, but again, is likely greatly influenced by reverse peristalsis in duodenum, affecting estimates in jejunum.

Yes, totally agree, this is the common disadvantage of estimating digestibility coefficients by using dietary marker. The current digestible amino acid system is also based on this method, yes, we agree the bigger variation in digestibility observed in the upper site of small intestine could be partially due to reverse peristalsis (and endogenous flow). Nevertheless, we still see correlations between performance and digestion rates in our previous reported studies.

1. Liu SY, Selle PH, Cowieson AJ. The kinetics of starch and nitrogen digestion regulate growth performance and nutrient utilisation in coarsely-ground, sorghum-based broiler diets Anim Prod Sci. 2013;53(10):1033-40.

2. Liu SY, Selle PH. A consideration of starch and protein digestive dynamics in chicken-meat production. World Poultry Sci J. 2015;71:297-310.

3. Per previous comment, this paper is very similar to the work conducted by Bryan et al. 2019 so the advancements, similarities, and differences between these 2 works should be addressed beyond the current text beginning in L396.

Respectfully, the Canadian group led by Prof Hank Classen is highly regarded and I look forward to reading their break-throughs in digestive dynamics in breeder nutrition. The present study was completed in 2016 but was delayed for publication due to the corresponding author’s maternity leave. The present study reported amino acid digestibility and CP digestibility in four sites of small intestine which is unique. The amino acid and protein digestibilities were only reported in the distal ileum by Bryan et al., 2019, it is unclear how the AA and protein digestion rates were derived in Bryan et al., 2019 without reporting AA digestibility in jejunums. One can only speculate they may have come across the same challenge, as indicated by the current reviewer, bigger variations in upper site of small intestine. However, it is important to report four sites digestibility to allow audience critically evaluate the concept and recent development of nutrient digestive dynamics.

4. There are likely sampling location and ingredient-dependent effects (e.g., fiber and mucin-associated AA thr, glyc, etc per discussion in L425-429) on endogenous AA losses, which were not accounted for in the experiment or considered in the interpretation. This should be addressed.

a. Were digesta squeezed or flushed to minimize epithelial sloughing/endogenous losses?

Intestinal segments were gently squeezed three times to minimise endogenous loss (included in M&M now). Flushing introduces far too much water and freeze-drying takes time and resources to go through 300+ samples (four sites).

5. Given the amount and complexity of the data, there seems to be quite a bit of discussion around secondary points that could be replaced with more directly relevant topics. Examples include L404 (processing effects on Lys dig); L434 to L449 (growth performance); L450 to 462 (energy utilization per comment above).

We have included the above suggested discussion.

Additional comments

- It would be helpful to keep the order of treatments in the diet table similar to that presented for the data tables.

Corrected

- “Potential digestible protein” is not clearly defined – is this the same as the asymptote described in line 189?

Yes, and corrected in line 189

- Provide units for Table 4 “disappearance rates”

Included

- Why were plasma data omitted from tables after Table 6?

The below is included as footnote in Table 5,

2Broiler chickens offered diet 2 did not generate enough digesta content for completing amino acid analyses in Table 5-14

- Is Table 13 necessary? There are no differences and the SEM is larger than the means reported in many cases, indicated data not incredibly useful.

Deleted and relevant discussion has been taken out

---

## [Decision Letter · Decision Letter 1]

1 Sep 2020

Initial assessment of protein and amino acid digestive dynamics in protein-rich feedstuffs for broiler chickens

PONE-D-20-12708R1

Dear Dr. Liu,

We’re pleased to inform you that your manuscript has been judged scientifically suitable for publication and will be formally accepted for publication once it meets all outstanding technical requirements.

Kind regards,

Juan J Loor

Academic Editor

PLOS ONE

Additional Editor Comments (optional):

Reviewers' comments:

Reviewer's Responses to Questions

**Comments to the Author**

1. If the authors have adequately addressed your comments raised in a previous round of review and you feel that this manuscript is now acceptable for publication, you may indicate that here to bypass the “Comments to the Author” section, enter your conflict of interest statement in the “Confidential to Editor” section, and submit your "Accept" recommendation.

Reviewer #1: All comments have been addressed

Reviewer #2: All comments have been addressed

Reviewer #3: All comments have been addressed

2. Is the manuscript technically sound, and do the data support the conclusions?

Reviewer #1: Yes

Reviewer #2: Yes

Reviewer #3: Yes

3. Has the statistical analysis been performed appropriately and rigorously? 

Reviewer #1: Yes

Reviewer #2: Yes

Reviewer #3: Yes

4. Have the authors made all data underlying the findings in their manuscript fully available?

Reviewer #1: Yes

Reviewer #2: Yes

Reviewer #3: Yes

5. Is the manuscript presented in an intelligible fashion and written in standard English?

Reviewer #1: Yes

Reviewer #2: Yes

Reviewer #3: Yes

6. Review Comments to the Author

Reviewer #1: (No Response)

Reviewer #2: (No Response)

Reviewer #3: (No Response)

7. PLOS authors have the option to publish the peer review history of their article (what does this mean?). If published, this will include your full peer review and any attached files.

Reviewer #1: No

Reviewer #2: No

Reviewer #3: No

---

## [Editor Report · Acceptance letter]

10 Sep 2020

PONE-D-20-12708R1 

Initial assessment of protein and amino acid digestive dynamics in protein-rich feedstuffs for broiler chickens 

Dear Dr. Liu:

I'm pleased to inform you that your manuscript has been deemed suitable for publication in PLOS ONE. Congratulations! Your manuscript is now with our production department. 

Kind regards, 

on behalf of

Dr. Juan J Loor 

Academic Editor

PLOS ONE